# Rainfall and streamflow sensor network design: a review of applications, classification, and a proposed framework

Juan C. Chacon-Hurtado[1], Leonardo Alfonso[1], Dimitri P. Solomatine [1,2]

[1] Department of Integrated Water Systems and Governance, UNESCO-IHE, Institute for Water Education, Delft, the Netherlands.
[2] Water Resources Section, Delft University of Technology, the Netherlands.

**Abstract.** Sensors and sensor networks play an important role in decision-making related to water quality, operational streamflow forecasting, flood early warning systems and other areas. In this paper we review a number of existing applications and analyse a variety of evaluation and design procedures for sensor networks with respect to various criteria. Most of the existing approaches focus on maximising the observability and information content of a variable of interest. From the context of hydrological modelling only a few studies use the performance of the hydrological simulation in terms of output discharge as a design criteria. In addition to the review, we propose a framework for classifying the existing design methods, and a generalised procedure for an optimal network design in the context of rainfall-runoff hydrological modelling.

**Keywords**: Sensor network design, Surface hydrological modelling, Precipitation, Discharge, Review, Geostatistics, Information Theory, Expert Recommendations

## 1 Introduction

Optimal design of sensor networks is a key procedure for improved water management as it provides information about the states of water systems. As the processes taking place in catchments are complex and the measurements are limited, the design of sensor networks is (and has been) a relevant topic since the beginning of the International Hydrological decade (1965 – 1974, TNO 1986) until today (Pham and Tsai 2016). During this period, the scientific community has not yet arrived to an agreement about a unified methodology for sensor network design due to the diversity of cases, criteria, assumptions, and limitations. This is evident from the range of existing reviews on hydrometric network design, such as those presented by WMO (1972), TNO (1986), Nemec and Askew (1986), Knapp and Marcus (2003), Pryce (2004), NRC (2004) and Mishra and Coulibaly (2009).

The design of rainfall and streamflow sensor networks depends to a large extent on the scale of the processes to be monitored and the objectives to address (TNO 1986, Loucks et al. 2005). Therefore, the temporal and spatial resolution of measurements are driven by the measurement objectives. For example, information for long-term planning does not require the same level of temporal resolution as for operational hydrology (WMO 2009, Dent 2012). On the global and country scale, sensor networks are commonly used for climate studies and trend detection (Cihlar et al. 2000, Grabs and Thomas 2002, WMO 2009, Environment Canada 2010, Marsh 2010, Whitfield et al. 2012), and denoted as National Climate Reference Networks (WMO 2009). On a regional or catchment-scale, applications require careful selection of monitoring stations, since water resources planning and

management decisions, such as operational hydrology and water allocation, require high temporal and spatial
resolution data (Dent 2012).

This paper presents a review of methods for optimal design and evaluation of precipitation and discharge sensor
networks at catchment scale, proposes a framework for classifying the design methods, and suggests a generalised
framework for optimal network design for surface hydrological modelling. It is possible to extend this framework
to other variables in the hydrological cycle, since optimal sensor location problems are similar. The framework
here introduced is part of the results of the FP7 WeSenseIt project (www.wesenseit.eu), and the validation of the
proposed methodology will be presented in subsequent publications. This review does not consider in-situ
installation requirements or recommendations, so the reader is referred to WMO (2008a) for the relevant and
widely accepted guidelines, and to Dent (2012) for current issues in practice.

The structure of this paper is as follows: first, a classification of sensor network design approaches according to
the explicit use of measurements and models is presented, including a review of existing studies. Next, a second
way of classification is suggested, which is based on the classes of methods for sensor network analysis, including
statistics, Information Theory, case-specific recommendations and others. Then, based on the reviewed literature,
an aggregation of approaches and classes is presented, identifying potential opportunities for improvement.
Finally, a general procedure for the optimal design of sensor networks is proposed, followed by conclusions and
recommendations.

### 1.1 Main principles of network design

The design of a sensor network use the same concepts as experimental design (Kiefer and Wolfowitz 1959, Fisher
1974). The design should ensure that the data is sufficient and representative, and can be used to derive the
conclusions required from the measurements. (EPA 2002), or to assess the water status of a river system (EC 2000).
In the context of rainfall-runoff hydrological modelling, provide the sufficient data for accurate simulation and
forecasting of discharge and water levels, at stations of interest.

The objectives of the sensor network design have been categorised into two groups, the optimality alphabet
(Fedorov 1972, Box 1982, Fedorov and Hackl 1997, Pukelsheim 2006, Montgomery 2012), which uses different
letters to name different design criteria, and the Bayesian framework (Chaloner en Verdinelli 1995, DasGupta
1996). The alphabetic design is based on the linearization of models, optimising particular criteria of the
information matrix (Fedorov and Hackl 1997). Bayesian methods are centred on principles of decision making
under uncertainty, in which it seeks to maximise the gain in Information (Shannon 1948) between the prior and
posterior distributions of parameters, inputs or outputs (Lindley 1956, Chaloner and Verdinelli 1995). Among the
most used alphabetic objectives are the D-optimal, which minimises the area of the uncertainty ellipsoids around
the model parameters; and G-optimal, which minimises the variance of the predicted variable, which can also be
used as objective functions in the Bayesian design.

These general objectives are indirectly addressed in the literature of optimisation of hydrometric sensor networks,
achieved by the use of several functional alternatives. These approaches do not consider block experimental design

(Kirk 2009), due to the incapacity to replicate initial conditions in a non-controlled environment, such as natural processes.

On the practical end, the design of a sensor network should start with the institutional setup, purposes, objectives and priorities of the network (Loucks et al. 2005, WMO 2008b). From the technical point of view, an optimal measurement strategy requires the identification of the process, for which data is required (Casman et al. 1988, Dent 2012). Considering that neither the information objectives are unique and consistent, nor the characterisation of the processes is complete, the re-evaluation of the sensor network design should occur on a regular basis. Therefore, the sensor network should be re-evaluated when either the studied process, information needs, information use, or the modelling objectives change. Consequently, regulations regarding monitoring activities are not often strict in terms of station density, but in the suitability of data to provide information about the status of the water system (EC 2000, EPA 2002).

The design of meteorological and hydrometric sensor networks should consider at least three aspects. First, it should meet various objectives that are sometimes conflicting (Loucks et al. 2005, Kollat et al. 2011). Second, it should be robust under the events of failure of one or more measurement stations (Kotecha et al. 2008). Third, it must take into account different purposes and users with different temporal and spatial scales (Singh et al. 1986). Therefore, the design of an optimal sensor network is a multi-objective problem (Alfonso et al. 2010b).

The sensor network design can also be seen from an economic perspective (Loucks et al. 2005). In most cases, the main limitation in the deployment of sensor networks is related to costs, being sometimes the main driver of decisions related to reduction of the monitoring networks. The valuation between the costs of the sensor networks and the cost of having insufficient information is not usually considered, because the assessment of the consequences of decisions is made a-posteriori (Loucks et al. 2005, Alfonso et al. 2016). In most studies, it is seen that the improvement of information content metrics (e.g., entropy, uncertainty reduction, among others) is marginal as the number of extra sensors increases (Pardo-Iguzquiza 1998, Dong et al. 2006, Ridolfi et al. 2011), and thus the selection of the adequate number of sensors can be based on a threshold in the rate of increment in the objective function. However, in many practical applications the number of available sensors may be defined by budget limitations. Therefore, the optimal number of sensors in a network is strictly case-specific (WMO 2008).

**1.2 Scenarios for sensor network design: Augmentation, relocation and reduction**

Scenarios for designing of sensor networks may be categorised into three groups: augmentation, relocation and reduction (NRC 2004, Mishra and Coulibaly 2009, Barca et al. 2015). *Augmentation* refers to the deployment of at least one additional sensor in the network, whereas *Reduction* refers to the opposite case, where at least one sensor is removed from the original network. *Relocation* is about repositioning the existing network nodes.

The lack of data usually drives the sensor network augmentation, whereas economic limitations usually push for reduction. These costs of the sensor network usually relate to the deployment of physical sensors in the field, transmission, maintenance and continuous validation of data (WMO 2008).

Augmentation and relocation problems are fundamentally similar, as they require estimation of the measured
variable at ungauged locations. For this purpose, statistical models of the measured variable are often employed.
For example, Rodriguez-Iturbe and Mejia (1974) described rainfall regarding its correlation structure in time and
space; Pardo-Igúzquiza (1998) expressed areal averages of rainfall events with ordinary Kriging estimation;
Chacón-Hurtado et al. (2009) represented rainfall fields using block Kriging. In contrast, for network reduction,
the analysis is driven by what-if scenarios, as the measurements become available. Dong et al. (2005) employ this
approach to re-evaluate the efficiency of a river basin network based on the results of hydrological modelling.
In principle, augmentation and relocation aim to increase the performance of the network (Pardo-Igúzquiza 1998,
Nowak et al. 2010). In reduction, on the contrary, network performance is usually decreased. The driver for these
decisions is usually related to factors such as operation and maintenance costs (Moss et al. 1982, Dong et al. 2005).
**1.3 Role of measurements in rainfall-runoff modelling**
The typical data flow for hydrological rainfall-runoff modelling can be summarised as in Fig. 1. For discharge
simulation, precipitation and evapotranspiration are the most common data requirements (WMO 2008, Beven
2012), while discharge data is commonly employed for model calibration, correction and update (Sun et al. 2015).
Data-driven hydrological models may use measured discharge as input variables as well (e.g., Solomatine and Xue
2004, Shrestha and Solomatine 2006). Methods for updating of hydrological models have been widely used in
discharge forecasting as data assimilation, using the model error to update the model states. In this way, more
accurate discharge estimates can be obtained (Liu et al. 2012, Lahoz and Schneider 2014). In real-time error
correction schemes, typically, a data-driven model of the error is employed which may require as input any of the
mentioned variables (Xiong and O'Connor 2002, Solomatine and Ostfeld 2008).
In a conceptual way, we can express the quantification of discharge at a given station as (Solomatine and Wagener
138 2011):

$$Q = \hat{Q}(x, \theta) + \varepsilon \qquad (\textbf{1})$$

Where $Q$ is the recorded discharge, $\hat{Q}(x, \theta)$ represents a hydrological model, which is function of measured
variables (mainly precipitation and discharge, $x$) and the model parameters ($\theta$). $\varepsilon$ is the simulation error, which is
ideally independent of the model, but in practice is conditioned by it. Considering that neither the measurements
are perfect, nor the model unbiased, the variance of the estimates is proportional to the uncertainty in the model
inputs, $\sigma^2(x)$, and the uncertainty in model parameters, $\sigma^2(\theta)$:

$$\sigma^2\left(\hat{Q}(x, \theta)\right) \alpha\ \sigma^2(x), \sigma^2(\theta) \qquad (\textbf{2})$$

**2 Classification of approaches for sensor network evaluation**

There is a variety of approaches for the evaluation of sensor networks, ranging from theoretically sound to more pragmatic. In this section, we provide a general classification of these approaches, and more details of each method are given in the next section.

Although most of the approaches for the design of sensor networks make use of data, some rely solely on experience and recommendations. Therefore, a first tier in the proposed classification consists of recognising both measurement-based and measurement-free approaches (Fig. 2). The former make use of the measured data to evaluate the performance of the network (Tarboton et al. 1987, Anctil et al. 2006), while the latter use other data sources (Moss and Tasker 1991), such as topography and land use.

**2.1 Measurement-based evaluation**

The measurement-based approach can be furtherly subdivided into model-free and model-based approaches (Fig. 2), depending on the use of modelling results in the performance metric.

**2.1.1 Model-free performance evaluation**

In model-free approaches, water systems and the external processes that drive their behaviour are observed through existing measurements, without the use of catchment models. Then, metrics about amount and quality of information in space and time are evaluated with regards to the management objectives and the decisions to be made in the system. Some performance metrics in this category are joint entropy (Krstanovic and Singh 1992), Information Transfer (Yang and Burn 1994), interpolation variance (Pardo-Igúzquiza 1998, Cheng et al. 2007) and autocorrelation (Moss and Karlinger 1974), among others. Fig. 3 presents the flowchart for the case when precipitation and discharge, as main drivers of catchment hydrology (WMO 2008) are considered, in model-free network evaluation.

Fundamentally, the model-free approach aims to minimise the variance of the measured variable, therefore, (and in theory) minimising the variance in the estimation (equation 3). However, a design that is optimal for estimation is not necessarily also optimal for prediction (Chaloner and Verdinelli 1995).

$$\min \sigma^2 \left( \hat{Q}(x, \theta) \right) \; \alpha \min \left( \sigma^2(x) \right) \tag{3}$$

Application of model-free approaches can be found in Krstanovic and Singh (1992), Nowak et al. (2010), Li et al. (2012). Model-free evaluations are suitable for sensor network design aiming mainly to water resources planning, in which diverse water interests must be balanced. Due to the lack of a quantitative performance metric that relates simulated discharge, this kind of evaluations do not necessarily improve rainfall-runoff simulations.

**2.1.2 Model-based performance evaluation**

In the model-based approach, the performance of sensor networks is carried out using a catchment model (Dong et al. 2005, Xu et al. 2013), In this case, measurements of precipitation are used to simulate discharge, which is

compared to the discharge measurements at specific locations. Therefore, any metric of the modelling error could
be used to evaluate the performance of the network. Fig. 4 presents a generic model-based approach for evaluating
sensor networks.

In the model-based design of sensor networks, it is assumed that the model structure and parameters are adequate.
Therefore, it is possible to identify a set of measurements ($x$) which minimise the modelling error as.

$$\min \sigma^2(\epsilon) \; \alpha \min\left(\left|Q - \hat{Q}(x, \theta)\right|\right) \tag{4}$$


The need for the catchment model and possible high computational efforts for multiple model runs are some
disadvantages of this approach. The computational load is especially critical in case of complex distributed models.
It is worth mentioning particular model error metrics (Nash and Sutcliffe 1970, Gupta et al. 2009) may qualify the
network by its ability to capture certain hydrological processes (Bennet et al. 2013), affecting the network
evaluation.

### 2.2 Measurement-free evaluation

As it is seen from its name, this approach does not require the previous collection of data of the measured variable
to evaluate the sensor network performance. The evaluation of sensor networks is based on either experience or
physical characteristics of the area such as land use, slope or geology. In this group of methods, the following can
be mentioned: case-specific recommendations (Bleasdale 1965, Wahl and Crippen 1984, Karasseff 1986, WMO
2008a) and physiographic components (Tasker 1986, Laize 2004). This approach is the first step towards any
sensor network development (Bleasdale 1965, Moss et al. 1982, Nemec and Askew 1986, Karasseff 1986).

### 3 Classification of methods for sensor network evaluation

In this section, we classify the methods used to quantify the performance of the sensor networks based on the
mathematical apparatus used to evaluate the network performance. These methods can be broadly categorised in
statistics-based, information theory-based, expert recommendations, and others.

### 3.1 Statistics-based methods

Statistics-based methods refer to methods where the performance of the network is evaluated with statistical
uncertainty metrics of the measured or simulated variable. These methods aim to minimise either interpolation
variance (Rodriguez-Iturbe and Mejia 1974, Bastin et al. 1984, Bastin and Gevers 1985, Pardo-Iguzquiza 1998,
Bonaccorso 2003), cross-correlation (Maddock 1974, Moss and Karlinger 1974, Tasker 1986), or model error
(Dong et al. 2005, Xu et al. 2015).

### 3.1.1 Interpolation variance (geostatistical)

Methods to evaluate sensor networks considering a reduction in the interpolation variance assume that for a
network to be optimal, the measured variable should be as certain as possible in the domain of the problem. To
achieve this, a stochastic interpolation model that provides uncertainty metrics is required. Geostatistical methods
such as Kriging (Journel and Huijbregts 1978, Cressie 1993), or Copula interpolation (Bárdossy 2006) have an
explicit estimation of the interpolation error. This characteristic makes it suitable to identify areas with expected
poor interpolation results, (Bastin et al. 1984, Pardo-Igúzquiza 1998, Grimes et al. 1999, Bonaccorso et al. 2003,
Cheng et al. 2007, Nowak et al. 2009, 2010, Shafiei et al. 2013).

In the case of Kriging, the optimal estimation of a variable at ungauged locations is assumed to be a linear
combination of the measurements, with a Gaussian distributed probability distribution function. Under the ordinary
Kriging formulation, the variance in the estimation ($\sigma^2$) of a variable at location ($u$) over a catchment is:

$$\sigma^2(u) = C_0 - \sum_{\alpha=1}^{n} \lambda_\alpha(u) - C(u_\alpha - u) \tag{5}$$


Where $C_0$ refers to the variance of the random field, $\lambda_\alpha$ are the Kriging weights for the station $\alpha$ at the ungauged
location $u$. $C(u_\alpha - u)$ is the covariance between the station $\alpha$ at the location $u_\alpha$ and the interpolation target at the
location $u$. $n$ represents the total number of stations in the neighbourhood of $u$ and used in the interpolation.

Therefore, as an objective function the optimal sensor network is such that the total Kriging variance (TKV) is
minimum:

$$TKV = \sum_{u=1}^{U} \sigma^2(u) \tag{6}$$


Where $U$ is the total number of discrete interpolation targets in the catchment or domain of the problem.

Bastin and Gevers (1984) optimised a precipitation sensor network at pre-defined locations to estimate the average
precipitation for a given catchment. Their selection of the optimal sensor location consisted of minimising the
normalised uncertainty by reducing the network. The main drawback of their approach is that the network can only
be reduced and not augmented. Similar approaches have also been used by Rodriguez-Iturbe and Mejia (1974),
Bogárdi et al. 1985, and Morrissey et al. (1995). Pardo-Igúzquiza (1998) advanced this formulation by removing
the pre-defined set of locations (allowing augmentation). Instead, rain gauges were allowed to be placed anywhere
in the catchment and its surroundings. A simulated annealing algorithm is used to search for the find the optimal
set of sensors to minimise the interpolation uncertainty.

Copula interpolation is a geostatistical alternative to Kriging for the modelling of spatially distributed processes
(Bárdossy 2006, Bárdossy and Li 2008, Bárdossy and Pegram 2009). As a geostatistical model, the copula provides
metrics of the interpolation uncertainty, considering not only the location of the stations and the model
parameterisation but also the value of the observations. Li et al. (2011) use the concept of copula to provide a
framework for the design of a monitoring network for groundwater parameter estimation, using a utility function,
related to the cost of a given decision with the available information.

In the case of copula, the full conditional probability distribution function of the variable is interpolated. As such, the interpolation uncertainty depends on the confidence interval, measured values, parameterisation of the copula and the relative position of the sensors in the domain of the catchment. More details on the formulation of copula-based design can be found in Bárdossy and Li (2008).

Cheng et al. (2007), as well as Shafiei et al. (2013), recognised that the temporal resolution of the measurements affects the definition of optimality in minimum interpolation variance methods. This change in the spatial correlation structure occurs due to more correlated precipitation data between stations in coarser sampling resolutions (Ciach and Krajewski 2006). For this purpose, the sensor network has to be split into two parts, a base network and non-base sensors. The former should remain in the same position for long periods, to characterise longer fluctuation phenomena, based on the definition of a minimum threshold for an area with acceptable accuracy. The latter is relocated to improve the accuracy of the whole system, and should be relocated as they do not provide a significant contribution to the monitoring objective.

Recent efforts have used minimum interpolation variance approaches to consider the non-stationarity assumption of most geostatistical applications in sensor network design (Chacon-Hurtado et al. 2014). To this end, changes in the precipitation pattern and its effect on the uncertainty estimation were considered during the development of a rainfall event.

**3.1.2 Cross-correlation**

The objective of minimum cross-correlation methods is to avoid placing sensors at sites that may produce redundant information. Cross-correlation was suggested by Maddock (1974) for sensor network reduction, as a way to identify redundant sensors. In this scope, the objective function can be written as:

$$\rho\left(X_i, X_j\right) = \sum_{i=1}^{n} \sum_{j=i+1}^{n} \frac{cov(x_i, x_j)}{\sigma(x_i)\sigma(x_j)} \tag{7}$$

Where *cov* is the covariance function between a pair of stations (*i, j*), and σ is the standard deviation of the observations.

Stedinger and Tasker (1985) introduced the method called Network Analysis Using Generalized Least Squares (NAUGLS), which assesses the parameters of a regression model for daily discharge simulation based on the physiographic characteristics of a catchment (Stedinger and Tasker 1985, Tasker 1986, Moss and Tasker 1991). The method builds a Generalised-Least-Square (GLS) covariance matrix of regression errors to correlate flow records and to consider flow records of different length, so the sampling mean squared error can be expressed as:

$$SMSE = \frac{1}{n} \sum_{i=1}^{j} X_i^T (X^T \Lambda^{-1} X)^{-1} X_i \tag{8}$$

Where $X [k, w]$ is the matrix of the ($k$) basin characteristics in a window of size $w$ at discharge measuring site $i$. $\Lambda$
is the GLS Weighting matrix, using a set of $n$ gauges (Tasker 1986)
A comparable method was proposed by Burn and Goulter (1991), who used a correlation metric to cluster similar
stations. Vivekanandan and Jagtap (2012) proposed an alternative for the location of discharge sensors in a
recurrent approach, in which the most redundant stations were removed, and the most informative stations
remained using the Cooks' D metrics, a measure of how the spatial regression model at a particular site is affected
by removing another station. The result of these type of sensors is sparse, as the redundancy of two sensors
increases with the inverse of the distance between them (Mishra and Coulibaly 2009).

**3.1.3 Model output error**

These methods assume that the optimal sensor network configuration is such that satisfy a particular modelling
purpose, e.g. a minimum error in simulated discharge. Considering this, the design of a sensor network should be
such that minimises the difference between the simulated and recorded variable:

$$\min f\left(\left|Q - \hat{Q}(x, \theta)\right|\right) \tag{9}$$

Where $f$ is a metric that summarises the vector error such as Bias, Root Mean Squared Error (RMSE), or Nash-
Sutcliffe Efficiency (NSE); $Q$ is the measurements of the simulated variable, and $\hat{Q}$ is the simulation results using
inputs $x$, and parameters $\theta$. Bias measures the mean deviation of the results between the observations ($Q$) and
simulation results ($\hat{Q}$) for $t$ pairs of observations and simulation results:

$$Bias = \frac{1}{n}\sum_{i=1}^{t}\left(\hat{Q}_i - Q_i\right) \tag{10}$$

This metric theoretically varies from minus infinity to infinity, and its optimal value is equal to zero. The root
mean square error (RMSE) measures the standard deviation of the residuals as:

$$RMSE = \sqrt{\frac{1}{n}\sum_{i=1}^{t}\left(\hat{Q}_i - Q_i\right)^2} \tag{11}$$

The RMSE can vary then from zero to infinity, where zero represents a perfect fit between model results and
observations. As RMSE is a statistical moment of the residuals, the result is a magnitude rather than a score.
Therefore, benchmarking between different case studies is not trivial. To overcome this issue, Nash and Sutcliffe
(1970) proposed a score (also known as coefficient of determination) based on the ratio of the model results in
variance over the observation variance as:

$$NSE = 1 - \frac{\sum_{i=1}^{t}(\hat{Q}_i - Q_i)^2}{\sum_{i=1}^{t}(Q_i - \overline{Q}_i)^2}$$ ( 12)


In which $Q$ are the measurements, $\hat{Q}$ are the model results and $\overline{Q}$ is the average of the recorded series.

Theoretically, this score varies from minus infinity to one. However, its practical range lies between zero and one.
On the one hand, an NSE equal to zero indicates that the model has the same explanatory capabilities that the mean
of the observations. On the other end, a value of one represents a perfect fit between model results and observations.
Model output error formulations have been used to identify the most convenient set of sensors that provide the
best model performance (Tarboton et al. 1987) to propose measurement strategies regarding the number of gauges
and sampling frequency.

Another application is provided by Dong et al. (2005) who proposed to evaluate the rainfall network using a
lumped HBV model. They found that the model performance does not necessarily improve when extra rain gauges
are placed. A similar approach was presented by Xu et al. (2013) who evaluated the effect of diverse rain gauge
locations on runoff simulation using a similar hydrological model. It was found that rain gauge locations could
have a significant impact and suggest that a gauge density less than 0.4 stations per 1000 km2 can negatively affect
the model performance.

Anctil et al. (2006) aimed at improving lumped neural network rainfall-runoff forecasting models through mean
areal rainfall optimisation, and concluded that different combinations of sensors lead to noticeable streamflow
forecasting improvements. Studies in other fields have also used this method. For example, Melles et al. (2009,
2011), obtained optimal monitoring designs for radiation monitoring networks, which minimise the prediction
error of mean annual background radiation. The main drawback of this approach is that multiple error metrics are
considered, as specific objectives relate to different processes

**3.2 Information Theory-based methods**
The use of Information Theory (Shannon 1948) in the design of sensor networks for environmental monitoring is
based on Communication Theory, which studies the problem of transmitting signals from a source to a receiver
throughout a noisy medium. Information Theory provides the possibility of estimating probability distribution
functions in the presence of partial information with the less biassed estimation (Jaynes 1957). Some of its concepts
are analogous to statistics concepts, and therefore similarities between entropy and uncertainty, as mutual
information and correlation, etc., can be found (Cover and Thomas 2005, Alfonso 2010, Singh 2013).

Information Theory-based methods for designing sensor networks mainly consider the maximisation of
information content that sensors can provide, in combination with the minimisation of redundancy among them
(Krstanovic and Singh 1992, Mogheir and Singh 2002, Alfonso et al. 2010a,b, Alfonso 2010, Alfonso et al. 2013,
Singh 2013). Redundancy can be measured by using either Mutual Information (Singh 2000, Steuer et al. 2002),
Directional Information Transfer (Yang and Burn 1994), Total Correlation (Alfonso et al. 2010a,b, Fahle et al.
2015), among others.

### 3.2.1 Entropy

The Principle of Maximum Entropy (POME) is based on the premise that probability distribution with the largest
remaining uncertainty (i.e., the maximum entropy) is the one that best represent the current stage of knowledge.
POME has been used as a criterion for the design of sensor networks, by allowing the identification of the set of
sensors that maximises the joint entropy among measurements (Krstanovic and Singh 1992). In other words, to
provide as much information content, from the Information Theory perspective, as possible (Jaynes 1988).
In the design of sensor networks, the objective is to maximise the joint entropy ($H$) of the sensor network as:

$$H(X_1, X_2, \dots, X_n) = -\sum_{i=1}^{k} \dots \sum_{j=1}^{m} p(x_{i1}, \dots x_{jm}) \log p(x_{i1}, \dots x_{jm}) \qquad (13)$$


Where $p(X)$ is the probability of the random variable $X$ to take a discrete value $x_m$. As in many applications, $X$ is a
continuous variable which has to be discretised (quantised) into intervals ($k$, $m$) to calculate its entropy. The
probabilities are calculated following frequency analysis, such that the probability of a variable $X$ to take a value
in the interval $i, \dots, j$ which is defined by the number of times in which this value appear, divided by the complete
length of the dataset. When calculating the entropy of more than one variable simultaneously (joint entropy), joint
probabilities are used.
Krstanovich and Singh (1992) presented a concise work on rainfall network evaluation using entropy. They used
POME to obtain multivariate distributions to associate different dependencies between sensors, such as joint
information and shared information, which was used later either reduce the network (in the case of high
redundancy) or expand it (in the case of lack of common information).
Fuentes et al. (2007) proposed an entropy-utility criterion for environmental sampling, particularly suited for air-
pollution monitoring. This approach considers Bayesian optimal sub-networks using an entropy framework,
relying on the spatial correlation model. An interesting contribution of this work is the assumption of non-
stationarity, contrary to traditional atmospheric studies, and relevant in the design of precipitation sensor networks.
The use of hydraulic 1D models and metrics of entropy have been used to select the adequate spacing between
sensors for water level in canals and polder systems (Alfonso et al. 2010a,b). This approach is based on the current
conditions of the system, which makes it useful for operational purposes, but it does not necessarily support the
modifications in the water system conditions or changes in the operation rules. Studies on the design of sensor
networks using these methods are on the rise in the last years (Alfonso 2010, Alfonso et al. 2013, Ridolfi et al.
2013, Banik et al 2017).

Benefits of POME include the robustness of the description of the posterior probability distribution since it aims
to define the less biassed outcome. This is because neither the models nor the measurements are completely certain.
Li et al. (2012) presented, as part of a multi-objective framework for sensor network optimisation, the criteria of
maximum (joint) entropy, as one of the objectives. Other studies in this direction have been presented by Lindley
(1956), Caselton and Zidek (1984), Guttorp et al. (1993), Zidek et al. (2000), Yeh et al. (2011) and Kang et al.
395     (2014).


More recently, Samuel et al. (2013) and Coulibaly and Samuel (2014), proposed a mixed method involving
regionalisation and dual entropy multi-objective optimisation (CRDEMO), which is a step forward if compared to
single-objective optimisation for sensor network design.

### 3.2.2 Mutual information (trans-information)

Mutual information is a measurement of the amount of information that a variable contains about another. This is
measured as the *relative entropy between the joint distribution and the product distribution* (Cover and Thomas
2005). In the simplest expression (two variables), the mutual information can be defined as:

$$I(X_1, X_2) = H(X_1) + H(X_2) - H(X_1, X_2) \tag{14}$$


where $H(X_1)$ and $H(X_2)$ is the entropy of each of the variables, and $H(X_1, X_2)$ is the joint entropy between them.
The extension of the mutual information for more than two variables should not only consider the joint entropy
between them, but also the joint entropy between pairs of variables, leading to a significantly complex expression
for the multivariate mutual information. Regarding this issue, the multivariate mutual information can be addressed
as a nested problem, such that:

$$I(X_1, X_2, \dots, X_n) = I(X_1, X_2, \dots, X_{n-1}) - I(X_1, X_2, \dots, X_{n-1}|X_n) \tag{15}$$


Where $I(X_1, X_2, \dots, X_n)$ is the multivariate mutual information among n variables, and $I(X_1, X_2, \dots, X_{n-1} | X_n)$ is the
conditional information of $n$-1 variables with respect to the $n^{\text{th}}$ variable. The conditional mutual information can
be understood as the amount of information that a set of variable share with another variable (or variables). The
conditional mutual information of two variables ($X_1$ and $X_2$) with respect to a third one ($X_3$) can be quantified as:

$$I(X_1, X_2|X_3) = H(X_1|X_3) - H(X_1|X_2, X_3) \tag{16}$$


Where $H(X_1 | X_3)$ is the conditional entropy of $X_1$ to $X_3$ and $H(X_1 | X_2, X_3)$ is the conditional entropy of $X_1$ with
respect to $X_2$ and $X_3$ simultaneously. The conditional entropy can be understood as the amount information that a
variable does not share with another. The joint entropy between two variables can be quantified as:

$$H(X_1|X_2) = \sum_{i=1}^{k} \sum_{j=1}^{m} p(X_{1i}, X_{2j}) \log \frac{p(X_{1i})}{p(X_{1i}, X_{2j})} \tag{17}$$


where $p(X_1, X_2)$ is the joint probability, for $k$ and $m$ discrete values, of $X_1$ and $X_2$.

An optimal sensor network should avoid collecting repetitive or redundant information, in other words, it should
be such that reduces the mutual (shared) information between sensors in the network. Alternatively, it should
maximise the transferred information from a measured to a modelled variable at a point of interest (Amorocho and
Espildora 1973). Following this idea, Husain (1987) suggested an optimisation scheme for the reduction of a rain
sensor network. His objective was to minimise the trans-information between pairs of stations. However,
assumptions of the probability and joint probability distribution functions are strong simplifications of this method.
To overcome these assumptions, the Directional Information Transfer (DIT) index was introduced (Yang and Burn
1994) as the inverse of the coefficient of non-transferred information (NTI) (Harmancioglu and Yevjevich 1985).
Both DIT and NTI are a normalised measure of information transfer between two variables ($X_1$ and $X_2$).

$$DIT = \frac{I(X_1, X_2)}{H(X_1)} \tag{18}$$


Particularly for the design of precipitation sensor networks, Ridolfi et al. (2011) presented a definition of the
maximum achievable information content for designing a dense network of precipitation sensors at different
temporal resolutions. The results of this study show that there exists a linear dependency between the non-
transferred information and the sampling frequency of the observations.

Total Correlation ($C$) is an alternative measure of the amount of shared information between two or more variables,
and has also been used as a measure of information redundancy in the design of sensor networks (Alfonso et al.
2010a, b, Leach et al. 2015) as:

$$C(X_1, \ldots, X_n) = \sum_{i=1}^{n} H(X_i) - H(X_1, \ldots, X_N) \tag{19}$$


Where $C(X_1, X_2, \ldots, X_n)$ is the total correlation among the $n$ variables, $H(X_i)$ is the entropy of the variable $i$, and
$H(X_1, X_2, \ldots, X_n)$ is the joint entropy of the $n$ variables. Total Correlation can be seen then as a simplification of
the multivariate mutual information, where only the interaction among all the variables is considered. In the design
of sensor networks, it is expected that the mutual information among the different variables is minimum, therefore,
the difference between the total correlation and multivariate mutual information tends to be minimised as well.
The advantage of total correlation is the computational advantage that represents assuming a marginal value for
the interaction among variables.

A method to estimate trans-information fields at ungauged locations has been proposed by Su and You (2014),
employing a trans-information-distance relationship. This method accounts for spatial distribution of precipitation,
supporting the augmentation problem in the design of precipitation sensor networks. However, as the relationship

between trans-information between sensors and their distance is monotonic, the resulting sensor networks are generally sparse.

### 3.3 Methods based on expert recommendations

### 3.3.1 Physiographic components

Among the most used planning tools for hydrometric network design are the technical reports presented by the WMO (2008), in which a minimum density of stations depending on different physiographic units, are suggested (Table 1). Although these guidelines do not provide an indication about where to place hydrometric sensors, rather they recommend that their distribution should be as uniform as possible and that network expansion has to be considered. The document also encourages the use of computationally aided design and evaluation of a more comprehensive design. For instance, Coulibaly et al. (2013) suggested the use of these guidelines to evaluate the Canadian national hydrometric network.

Moss et al. (1982) presented one of the first attempts to use physiographic components in the design of sensor networks in a method called Network Analysis for Regional Information (NARI). This method is based on relations of basin characteristics proposed by Benson and Matalas (1967). NARI can be used to formulate the following objectives for network design: minimum cost network, maximum information and maximum net benefit from the data-collection program, in a Bayesian framework, which can be approximated as:

$$\log \sigma \big( S(|\hat{Q} - Q|)^{\alpha} \big) = a + \frac{b_1}{n} + \frac{b_2}{y} \tag{20}$$

where the function $S(|\hat{Q} - Q|)^{\alpha}$ is the $\alpha$ percentile of the standard error in the estimation of $Q$, $a$, $b_1$ and $b_2$ are the parameters from the NARI analysis, $n$ is the number of stations used in the regional analysis, and $y$ is the harmonic mean of the records used in the regression.

Laize (2004) presented an alternative for evaluating precipitation networks based on the use of the Representative Catchment Index (RCI), a measure to estimate how representative a given station in a catchment is for a given area, on the stations in the surrounding catchments. The author argues that the method, which uses datasets of land use and elevation as physiographical components, can help identifying areas with a insufficient number of representative stations on a catchment.

### 3.3.2 Practical case-specific considerations

Most of the first sensor networks were designed based on expert judgement and practical considerations. Aspects such as the objective of the measurement, security and accessibility are decisive to select the location of a sensor. Nemec and Askew (1986) presented a short review of the history and development of the early sensor networks, where it is highlighted that the use of "basic pragmatic approaches" still had most of the attention, due to its practicality in the field and its closeness with decision makers.

Bleasdale (1965) presented a historical review of the early development process of the rainfall sensor networks in the United Kingdom. In the early stages of the development of precipitation sensor networks, two main characteristics influencing the location of the sensors were identified: at sites that were conventionally satisfactory and where good observers were located. However, the necessity of a more structured approach to select the location of sensors was underlined. As a guide, Bleasdale (1965) presented a series of recommendations on the minimal density of sensors for operational purposes, summarised in Fig. 5, relating the characteristics of the area to be monitored and the minimum required a number of rain sensors, as well as its temporal resolution.

In a more structured approach, Karasseff (1986) introduced some guidelines for the definition of the optimal sensor network to measure hydrological variables for operational hydrological forecasting systems. The study specified the minimum requirements for the density of measurement stations based on the fluctuation scale and the variability of the measured variable by defining zonal representative areas. This author suggested the following considerations for selecting the optimal placement of hydrometric stations:

- *"in the lower part of inflow and wastewater canals"*

- *"at the heads of irrigation and watering canals taking water from the sources"*

- *"at the beginning of a debris cone before the zone of infiltration, and at its end, where ground-water decrement takes place"*

- *"at the boundaries of irrigated areas and zones of considerable industrial water diversions (towns) "*

- *"at the sites of hydroelectric power plants and hydro projects"*

From a different perspective, Wahl and Crippen (1984), as well as Mades and Oberg (1986) proposed a qualitative score assessment of different factors related to the use of data and the historical availability of records for the evaluation of sensor value. Their analyses aimed at identifying candidate sensors to be discontinued, due to their limited accuracy.

### 3.3.3 User survey

These approaches aim to identify the information needs of particular groups of users (Sieber 1970), following the idea that the location of a certain sensor (or group of sensors) should satisfy at least one specific purpose. To this end, surveys to identify the interests for the measurement of certain variables, considering the location of the sensor, record length, frequency of the records, methods of transmission, among others, are executed.

Singh et al. (1986) applied two questionnaires to evaluate the streamflow network in Illinois: one to identify the main uses of streamflow data collected at gauging stations, where participants described how data was used and how they would categorise it in either site-specific management activities, local or regional planning and design, or determination of long-term trends. The second questionnaire was used to determine present and future needs for streamflow information. The results showed that the network was reduced due to the limited interest about

certain sensors, which allowed for enhancing the existing network using more sophisticated sensors or recording
methods. Additionally, this redirection of resources increased the coverage at specific locations.

**3.4 Other methods**

There are also other methods that cannot be easily attributed to the previously mentioned categories. Among them,
Value of Information, fractal, and network theory-based methods can be mentioned.

**3.4.1 Value of Information**

The Value of Information (VOI, Howard 1966, Hirshleifer and Riley 1979) is defined as the value a decision-
maker is willing to pay for extra information before making a decision. This willingness to pay is related to the
reduction of uncertainty about the consequences of making a wrong decision (Alfonso and Price 2012).
The main feature of this approach is the direct description of the benefits of additional piece of information,
compared with the costs of acquiring that extra piece of information (Black et al. 1999, Walker 2000, Nguyen and
Bagajewicz 2011, Alfonso and Price 2012, Ballari et al. 2012). The main advantage of this method is that provides
a pragmatic framework in which information have a utilitarian value, usually economic, which is especially suited
for budget constraint conditions.
One of the assumptions of this type of models is that a prior estimation of consequences is needed. If $a$ is the action
that has been decided to perform, $m$ is the additional information that comes to make such a decision, and $s$ is the
state that is actually observed, then the expected utility of any action a can be expressed as:

$$u(a, P_s) = \sum_S P_s u(C_{as})$$ ( 21)

where $P_s$ is the perception, in probabilistic terms, of the occurrence of a particular state ($s$) among a total number
of possible states ($S$), and $u$ is the utility of the outcome $C_{as}$ of the actions given the different states. When new
information (i.e., a message $m$) becomes available, and the decision-maker accepts it, his prior belief $P_s$ will be
subject to a Bayesian update. If $P(m|s)$ is the likelihood of receiving the message $m$ given the state $s$ and $P_m$ is the
probability of getting a message $m$ then:

$$P_m = \sum_S P_s P(m|s)$$ ( 22)

The value of a single message $m$ can be estimated as the difference between the utility, $u$, of the action, $a_m$ that is
chosen given a particular message $m$ and the utility of the action, $a_0$, that would have been chosen without
additional information as:

$$\Delta_m = u(a_m, P(s|m)) - u(a_0, P(s|m))$$ ( 23)


The Value of Information, *VOI*, is the expected utility of the values $\Delta_m$:

$$VOI = E(\Delta_m) = \sum_M P_m \Delta_m \qquad (24)$$


Following the same line of ideas, Khader et al. (2013) proposed the use of decision trees to account for the
development of a sensor network for water quality in drinking groundwater applications. VOI is a straightforward
methodology to establish present causes and consequences of scenarios with different types of actions, including
the expected effect of additional information. A recent effort by Alfonso et al. (2016) towards identifying valuable
areas to get information for floodplain planning consists of the generation of VOI maps, where probabilistic flood
maps and the consequences of urbanisation actions are taken into account to identify areas where extra information
may be more critical.
**3.4.2 Fractal-based**
Fractal-based methods employ the concept of Gaussian self-affinity, where sensor networks show the same spatial
patterns at different scales. This affinity can be measured by its fractal dimension (Mandelbrot 2001). Lovejoy et
al. (1986) proposed the use of fractal-based methods to measure the dimensional deficit between the observations
of a process and its real domain. Consider a set of evenly distributed cells representing the physical space, and the
fractal dimension of the network representing the number of observed cells in the correlation space. The lack of
non-measured cells in the correlation space is known as the fractal deficit of the network. Considering that a large
number of stations have to be available at different scales, the method is suitable for large networks, but less useful
in the deployment of few sensors in a catchment scale.

Lovejoy and Mandelbrot (1985) and Lovejoy and Schertzer (1985) introduced the use of fractals to model
precipitation. They argued that the intermittent nature of the atmosphere can be characterised by fractal measures
with fat-tailed probability distributions of the fluctuations, and stated that standard statistical methods are
inappropriate to describe this kind of variability. Mazzarella and Tranfaglia (2000) and Capecchi et al. (2012)
presented two different case studies using this method for the evaluation of a rainfall sensor networks. The former
study concludes that for network augmentation, it is important to select the optimal locations that improve the
coverage, measured by the reduction of the fractal deficit. However, there are no practical recommendations on
how to select such locations. The latter proposes the inspection of seasonal trends as the meteorological processes
of precipitation may have significant effects on the detectability capabilities of the network.

A common approach for the quantification of the dimensional deficit is the box-counting method (Song et al. 2007,
Kanevski 2008), mainly used in the fractal characterisation of precipitation sensor networks. The fractal dimension
of the network ($D$) is quantified as the ratio of the logarithm of the number of blocks ($NB$) that have measurements
and the logarithm of the scaling radius ($R$).

$$D = \frac{\log(NB(R))}{\log(R)} \qquad (25)$$


Due to the scarcity of measurements of precipitation type of networks, the quantification of the fractal dimension
may result unstable. An alternative fractal dimension may be calculated using a correlation integral (Mazzarella &
Tranfaglia 2000) instead of the number of blocks, such that:

$$CI(R) = \frac{2}{B(B-1)} \sum_{i=1}^{B} \sum_{j=1}^{B} \Theta\left(R - |u_{\alpha i} - u_{\alpha j}|\right) : for \; i \neq j \qquad (26)$$


In which $CI$ is the correlation integral, $R$ is the scaling radius, $B$ is the total number of blocks at each scaling radius,
and $U_\alpha$ is the location of station α. $\Theta$ is the Heaviside function. A normalisation coefficient is used, as the number
of estimations of the counting of blocks considers each station as a centre.

The consequent definition of the fractal dimension of the network is the rate between the logarithm of the
correlation integral and the logarithm of the scaling radius. This ratio is calculated from a regression between
different values of $R$, for which the network exhibit fractal behaviour (meaning, a high correlation between log($CI$)
and log($R$)).

$$D = \frac{\log(CI)}{\log(R)} \qquad (27)$$


The Maximum potential value for the fractal dimension of a 2-D network (such as for spatially distributed
variables) is two. However, this limit considers that the stations are located on a flat surface, as elevation is
consequence of the topography, and is not a variable that can be controlled in the network deployment.
**3.4.3 Network theory-based**
Recently, research efforts have been devoted to the use of the so-called network theory to assess the performance
of discharge sensor networks (Sivakumar and Woldemeskel 2014, Halverson and Fleming 2015). These studies
analyse three main features, namely average clustering coefficient, average path length and degree distribution.
Average clustering is a degree of the tendency of stations to form clusters. Average path length is the average of
the shortest paths between every combination of station pairs. Degree distribution is the probability distribution of
network degrees across all the stations, being network degree defined as the number of stations to which a station
is connected. Halverson and Fleming (2015) observed that regular streamflow networks are highly clustered (so
the removal of any randomly chosen node has little impact on the network performance) and have long average
path lengths (so information may not easily be propagated across the network).

In hydrometric networks, three metrics are identified (Halverson and Fleming 2015): degree distribution,
clustering coefficient and average path length. The first of these measures is the average node degree, which
corresponds to the probability of a node to be connected to other nodes. The metric is calculated in the adjacency
matrix (a binary matrix in which connected nodes are represented by 1 and the missing links by 0). Therefore, the
degree of the node is defined as:

$$k(\alpha) = \sum_{j=1}^{n} a_{\alpha,j} \qquad\qquad (28)$$


Where $k(\alpha)$ is the degree of station $\alpha$, $n$ is the total number of stations, and a is the adjacency matrix.

The clustering coefficient is a measure of how much the nodes cluster together. High clustering indicates that
nodes are highly interconnected. The clustering coefficient (*CC*) for a given station is defined as:

$$CC(\alpha) = \frac{2}{k(\alpha)(k(\alpha) - 1)} \sum_{j=1}^{n} a_{\alpha,j} \qquad\qquad (29)$$


Additionally, the average path length refers to the mean distance of the interconnected nodes. The length of the
connections in the network, provide some insights in the length of the relationships between the nodes in the
network.

$$L = \frac{1}{n(n-1)} \sum_{\alpha=1}^{k(\alpha)} \sum_{j=1}^{n} d_{\alpha,j} \qquad\qquad (30)$$


As can be seen from the formulation, the metrics of the network largely depends on the definition of the network
topology (adjacency matrix). The links are defined from a metric of statistical similitude such as the Pearson r or
the Spearman rank coefficient. The links are such pair of stations over which statistical similitude is over a certain
threshold.

According to Halverson and Fleming (2015), an optimal configuration of streamflow networks should consist of
measurements with small membership communities, high-betweenness, and index stations with large numbers of
intracommunity-links. Small communities represent clusters of observations, thus, indicating efficient
measurements. Large numbers of intra-community links ensure that the network has some degree of redundancy,
and thus, resistant to sensor failure. High-betweenness indicates that such stations which have the most inter-
communal links are adequately connected, and thus, able to capture the heterogeneity of the hydrological processes
at a larger scale.
**3.5 Aggregation of approaches and classes**
Table 2 summarises the sensor network design classes and approaches, with the selected references to the relevant
papers in each of the categories for further reference.

It is of special interest in the review to highlight the lack of model-based information theory methods, as well as
the little amount of publications in network theory-based methods. Also, quantitative studies in the comparison of
different methodologies for the design of sensor networks are limited. It is suggested, therefore, that a pilot
catchment is used for the scientific community to test all the available methods for network evaluation, establish
similarities and differences among them.
Table 3 summarises the main advantages and disadvantages for each of the design and evaluation methods. These
recommendations are general, but take into account the most general points in the design considerations of sensor
networks. Some of the advantages of these methods have been exploited in combined methodologies, such as those
presented by Yeh et al. (2011), Samuel et al. (2013), Barca et al. (2014), Coulibaly and Samuel (2014) and Kang
et al. (2014).
**4 General procedure for sensor network design**
Based on the presented literature review, in this section an attempt is made to present a first version of a unified,
general procedure for sensor network design. Such procedure logically link in a flowchart various methods,
following the measurement-based approaches (Fig. 6). The flowchart suggests two main loops: one to measure the
network performance (optimisation loop), and a second one to represent the selection in the number of sensors in
either augmentation or reduction scenarios. Most of the measurement-based methods, as well as most of the design
scenarios can be typically seen as particular cases of this generalised algorithmic flowchart.
The general procedure consists of 11 steps (boxes in Fig. 6). In the first place, physical measurements (1) are
acquired by the sensor network. This data is used to parameterise an estimator (2), which will be used to estimate
the variable at the Candidate Measurement Locations (CML) using, for instance, Kriging (Pardo-Igúzquiza 1998,
Nowak et al. 2009), or 1D hydrodynamic models (Neal et al. 2012, Rafiee 2012, Mazzoleni et al. 2015). The sensor
network reduction does not require such estimator as measurements are already in place.
The selection of the CML should consider factors such as physical and technical availability, as well as costs
related to maintenance and accessibility of stations, as illustrated by the WMO (2008) recommendations. The
selection of CML can also be based, for example, on expert judgement. These limitations may be presented in the
form of constraints in the optimisation problem.
Then an optimisation loop starts (Fig. 6), by the estimation of the measured variable at the CML (3), using the
estimator built in (2). Next, the performance of the sensor network at the CML is evaluated (4), using any of the
previously discussed methods. The selection of the method depends on the designer and its information
requirements, which also determines if an optimal solution is found (5). The stopping criteria in the optimisation
problem can be set by a desired accuracy of the network, some non-improved number of solutions or a maximum
number of iterations. As pointed out in the review, these performance metrics can be either model-based or model-
free and should not be confused with the use of a (geostatistical) model of the measured variable.
In case the optimisation loop is not complete, a new set of CML is selected (6). The use of optimisation algorithms
may drive the search of the new potential CML (Pardo-Igúzquiza 1998, Kollat et al. 2008, Alfonso 2010, Kollat
et al. 2011). The decision about adequate performance should not only consider the expected performance of the
network but also, recognise the effect of a limited number of sensors.

Once the performance is optimal, an iteration over the number of sensors is required. If the scenario is for network
augmentation (7), then a possibility of including additional sensors has to be considered (8). The decision to go
for an additional sensor will depend on the constraints of the problem, such as a limitation on the number of sensors
to install, or on the marginal improvement of performance metrics.

The network reduction scenario (9) is inverse: due to diverse reasons, mainly of financial nature, networks require
to have fewer sensors. Therefore, the analysis concerns what sensors to remove from the network, within the
problem constraints (10).

Finally, the sensor network is selected (11) from the results of the optimisation loop, with the adequate number of
sensors. It is worth mentioning that an extra loop is required, leading to re-evaluation, typically done on a periodical
basis, when objectives of the network may be redefined, new processes need to be monitored, or when information
from other sources is available, and that can potentially modify the definition of optimality.

**5 Conclusions and recommendations**
This paper summarises some of the methodological criteria for the design of sensor networks in the context of
hydrological modelling, proposed a framework for classifying the approaches in the existing literature and also
proposed a general procedure for sensor network design. The following conclusions can be drawn:

Most of the sensor network methodologies aim to minimise the uncertainty of the variable of interest at ungauged
locations and the way this uncertainty is estimated varies between different methods. In statistics-based models,
the objective is usually to minimise the overall uncertainty about precipitation fields or discharge modelling error.
Information theory-based methods aim to find measurements at locations with maximum information content and
minimum redundancy. In network theory-based methods, estimations are generally not accurate, resulting in less
biassed estimations. In methods based on practical case-specific considerations and value of information, the
critical consequences of decisions dictate the network configuration.

However, in spite of the underlying resemblances between methods, different formulations of the design problem
can lead to rather different solutions. This gap between methods has not been deeply covered in the literature and
therefore a general agreement on sensor network design procedure is relevant.

In particular, for catchment modelling, the driving criteria should also consider model performance. This driving
criterion ensures that the model adequately represents the states and processes of the catchment, reducing model
uncertainty and leading to more informed decisions. Currently, most of the network design methods do not ensure
minimum modelling error, as often it is not the main performance criteria for design.

Furthermore, in the last years, the rise of various sensing technologies in operational environments have promoted
the inclusion of additional design considerations towards a unified heterogeneous sensor network. These new
sensing technologies include, e.g., passive and active remote sensing using radars and satellites (Thenkabali 2015),
microwave link (Overeem et al. 2011), mobile sensors (Haberlandt and Sester 2010, Dahm et al. 2014),
crowdsourcing and citizen observatories (Huwald et al. 2013, Lanfranchi et al. 2014, Alfonso et al. 2015). These
non-conventional information sources have the potential to complement conventional networks, by exploiting the
synergies between the virtues and reducing limitations of various sensing techniques, and at the same time, require
the new network design methods allowing for handling the heterogeneous dynamic data with varying uncertainty.
The proposed classification of the available network design methods was used to develop a general framework for
network design. Different design scenarios, namely relocation, augmentation and reduction of networks are
included, for measurement-based methods. This framework is open and offers "placeholders" for various methods
to be used depending on the problem type.
Concerning the further research, from the hydrological modelling perspective, we propose to direct efforts towards
the joint design of precipitation and discharge sensor networks. Hydrological models use precipitation data to
provide discharge estimates, however as these simulations are error-prone, the assimilation of discharge data, or
error correction, reduces the systematic errors in the model results. The joint design of both precipitation and
discharge sensor networks may help to provide more reliable estimates of discharge at specific locations.
Another direction of research may include methods for designing dynamic sensor networks, given the increasing
availability of low-cost sensors, as well as the expansion of citizen-based data collection initiatives
(crowdsourcing). These information sources are on the rise in the last years, and one may foresee appearance of
interconnected, multi-sensor heterogeneous sensor networks shortly.
The presented review has also shown that limited effort has been devoted to considering changes in long-term
patterns of the measured variable in the sensor network design. This assumption of stationarity has become more
relevant in the last years due to new sensing technologies and increased systemic uncertainties, e.g. due to climate
and land use change and rapidly changing weather patterns. Although this topic has been recognised for quite some
time already (see e.g. Nemec and Askew 1986), the number of publications presenting effective methods to deal
with them is still limited. This problem, and the techniques to solve it, are being addressed in the ongoing research.

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

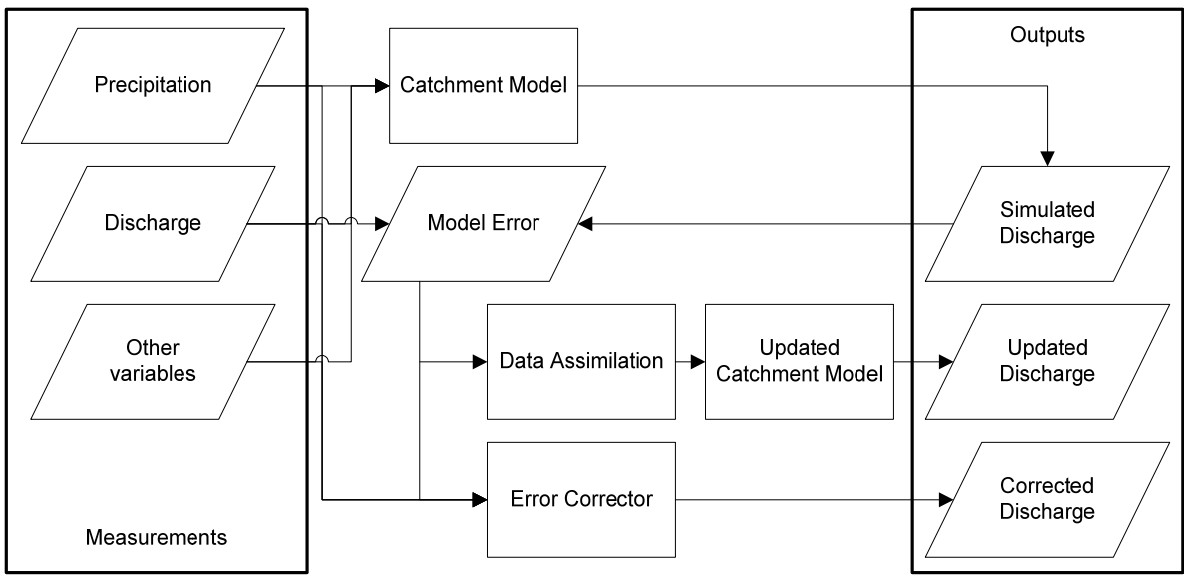


**Figure 1 Typical data flow in discharge simulation with hydrological models**

```
Approaches
┌─────────────────────┐  ┌─────────────────────┐
│ Measurement based   │  │  Measurement free   │
└─────────────────────┘  └─────────────────────┘

┌──────────────┐  ┌──────────────┐
│  Model-free  │  │ Model-based  │
└──────────────┘  └──────────────┘
```

Classes of methods

Statistics-based methods

Information Theory-based methods

Methods based on expert recommendations

Other methods


**Figure 2 Proposed classification of methods for sensor network evaluation**

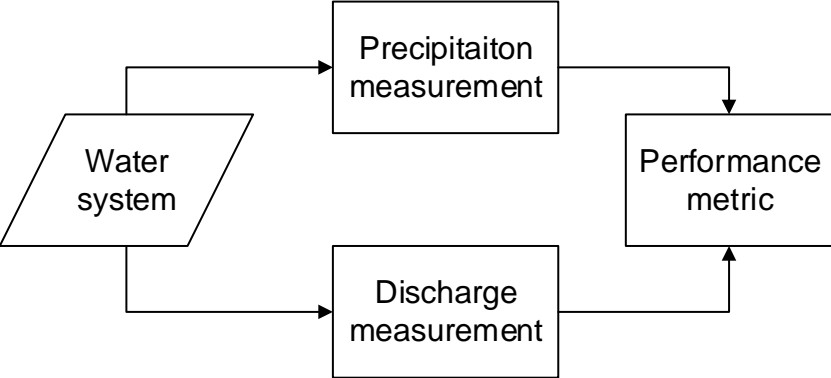


**Figure 3 General procedure for Model-free sensor network evaluation**

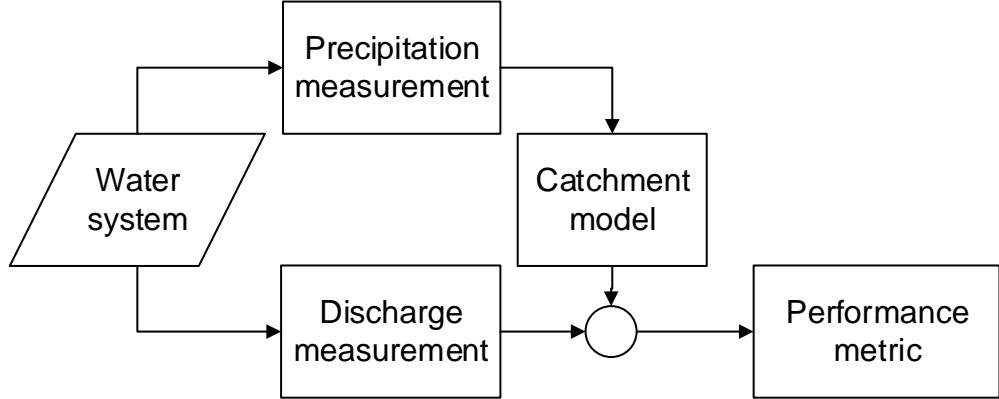


**Figure 4 General procedure for Model-based sensor network evaluation**

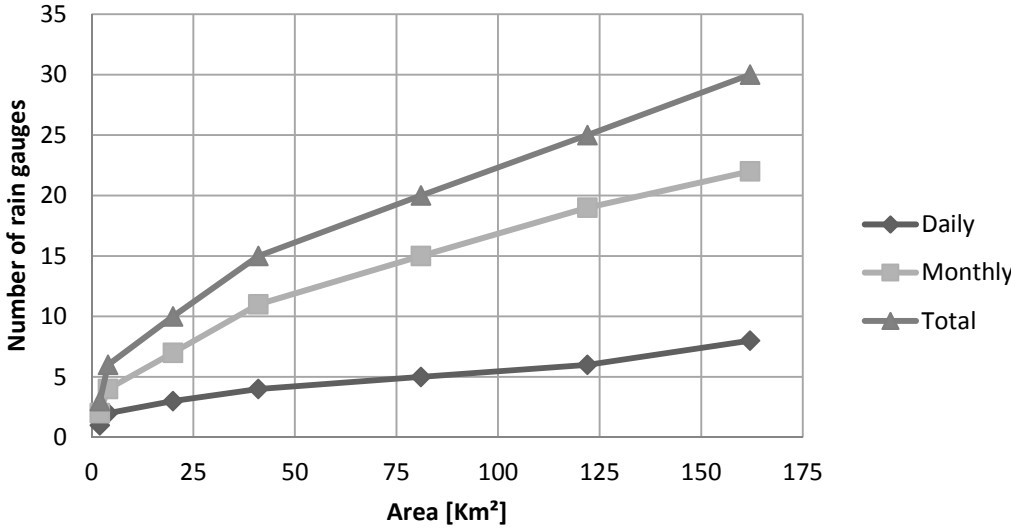


**Figure 5 Minimum number of rain gauges required in reservoired moorland areas - adapted from: (Bleasdale 1965)**

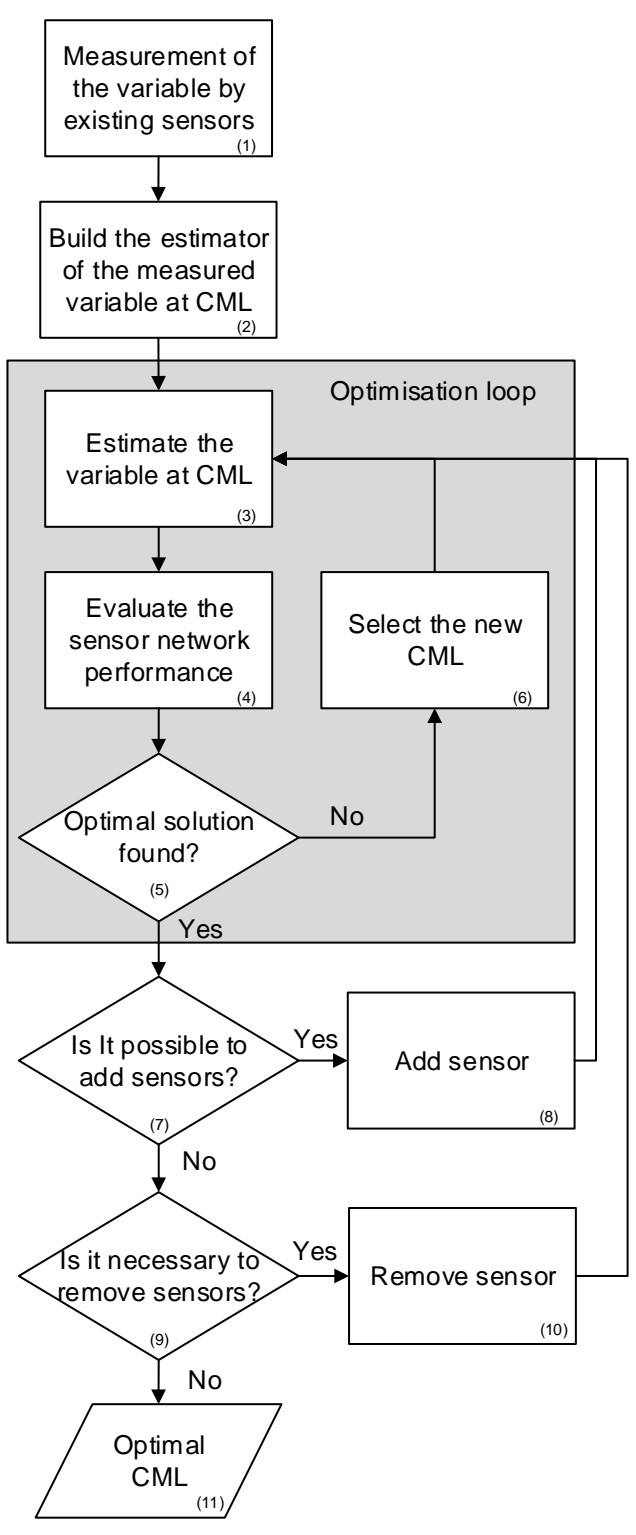


**Figure 6 Sensor network (re) design flow chart. (CML=candidate measurement locations)**

**Table 1 Recommended minimum densities of stations (area in Km² per station) – Adopted from WMO [2008]**

| Physiographic unit | Precipitation | | Evaporation | Streamflow | Sediments | Water Quality |
|---|---|---|---|---|---|---|
| | Non-recording | Recording | | | | |
| Coastal | 900 | 9,000 | 50,000 | 2,750 | 18,300 | 55,000 |
| Mountains | 250 | 2,500 | 50,000 | 1,000 | 6,700 | 20,000 |
| Interior plains | 575 | 5,750 | 5,000 | 1,875 | 12,500 | 37,500 |
| Hilly/undulating | 575 | 5,750 | 50,000 | 1,875 | 12,500 | 47,500 |
| Small islands | 25 | 250 | 50,000 | 300 | 2,000 | 6,000 |
| Urban areas | – | 10–20 | – | – | – | – |
| Polar/arid | 10,000 | 10,000 | 100,000 | 20,000 | 200,000 | 200,000 |



**Table 2 Classification of sensor network design criteria including recommended reading**

| | | Approaches | | |
|---|---|---|---|---|
| | | **Measurement-based** | | **Measurement-Free** |
| | | **Model-free** | **Model-based** | |
| | | **Statistics-based** | | |
| Classes | Interpolation variance | Pardo-Iguzquiza (1998) Bardossy and Li (2008) Nowak et al. (2010) | | |
| | Cross-correlation | Maddock (1974) Moss and Karlinger (1974) | Vivekanandan and Jagatp (2012) | |
| | Model error | | Tarboton et al. (1987) Dong et al. (2005) | |
| | | **Information Theory** | | |
| | Entropy | Krstanovic and Singh (1992) Alfonso et al. (2014) | Pham and Tsai (2016) | |
| | Mutual information | Husain (1987) Alfonso (2010) | Coulibaly and Samuel (2014) | |
| | | **Expert recommendations** | | |
| | Physiographic components | Samuel et al. (2013) | Moss and Karlinger (1974) Moss et al. (1982) | Lazie (2004) |
| | Practical case-specific considerations | | | Wahl and Crippen (1984) Nemec and Askew (1986) Karaseff (1986) |
| | User survey | | | Sieber (1970) Singh et al. (1986) |
| | | **Other methods** | | |
| | Value of information | Alfonso and Price (2012) | Black et al. (1999) Alfonso et al. (2016) | |
| | Fractal characterisation | | | Lovejoy and Mandelbrot (1985) Capecchi et al. (2012) |
| | Network theory | Sivakumar and Woldemeskel (2014) Halverson and Fleming (2015) | | |



**Table 3 Advantages and disadvantages of sensor network design methods**

| | Advantages | Disadvantages |
|---|---|---|
| **Statistics-based** | | |
| Interpolation variance | Useful to assess data scarce areas<br><br>No event-driven<br><br>Minimise uncertainty in spatial distribution of measured variable | Heavily rely on the characterisation of the covariance structure<br><br>No relationship with final measurement objective |
| Cross-correlation | Useful for detecting redundant stations<br><br>Computationally inexpensive | Augmentation not possible without additional assumptions<br><br>Limited to linear dependency between stations |
| Model error | Has direct relationship with the measurement objectives | Biased towards current measurement objectives<br><br>Biased towards model and error metrics |
| **Information Theory** | | |
| Entropy | Assess non-linear relationship between variables<br><br>Unbiased estimation of network performance | Formal form is computationally intensive<br><br>Quantising (binning) of continuous variables lead to different results<br><br>Optimal networks are usually sparse<br><br>Difficult to benchmark<br><br>Data intensive |
| Mutual information | Idem | Idem |
| **Expert recommendations** | | |
| Physiographic components | Reasonably well understood<br><br>Functional for heterogeneous catchments with few available measurements<br><br>Useful at country/continental level | Not useful for homogeneous catchments<br><br>No quantitative measure of network accuracy |
| Practical case-specific considerations | No previous measurements are required<br><br>Useful to observe specific variables | Biased towards expert<br><br>Collected data does not influence selection<br><br>Biased towards current data requirements |
| User survey | Pragmatic<br><br>Cost-efficient | Extensive user identification<br><br>Biased towards current data requirements |
| **Other methods** | | |
| Value of information | Provides assessment using economics concepts<br><br>Takes into account decision-maker's prior beliefs in the assessment | Consequences of decisions are difficult to quantify<br><br>Usually decisions are made with available information<br><br>Biased towards a rational decision model |
| Fractal characterisation | Efficient for large networks<br><br>Does not require data collection | Not suitable for small networks or catchments<br><br>Does not consider topographic or orographic influence |
| Network theory | Provides insight in interconnected networks | Not useful for augmentation purposes<br><br>Data intensive |