# Peer review of "Rainfall and streamflow sensor network design: a review of applications, classification, and a proposed framework"

_Hydrology and Earth System Sciences, 2016_

## Referee Comment (RC1) · Anonymous Referee #1 · 5 Oct 2016

The authors present an overall picture on hydrometric network design methods and approaches to increase or reduce sensor density using different mehods e.g. expert opinons and hydrologic models. They also classify these methods and present an optimal network design using complementary rainfall-runoff model performance. The use of hydrologic model makes sense as the products of the sensors are usually used by the hydrologic models. This review paper addresses an interesting topic. However, the presentation of the cases needs some more details on country scale applications as listed below. What are the practices in very densely monitored countries (e.g. Germany) and data scarce ones (e.g. Poland, Spain and Turkey). Also what is the optimum level of network density.

Overall, major revision is recommended for the the manuscript.

Specific Comments:

1. Title: Rainfall and streamflow sensor network design: a review of applications, classification, and a proposed framework

Recommended title: Review of precipitation and streamflow sensor network design methods from hydrologic modeling perspective

2. Section/subsection titles should be reorganized in a clear way. For example sub-section 3.3.2 Methods based on expert judgement and 3.3 Methods based on expert recommendations are similar and confusing.

3. In most of the European countries (e.g. Denmark and Germany) or even in USGS, the number of rainfall/streamflow sensors/stations is decreasing due to maintenance costs and use of radar data. I would expect to read some more insight on specific examples about sensor density and the country based approaches. Compare, for example, Spain/Poland and Germany from network density aspect to indicate an optimum approach. Now the content is very technical and dry for the reader.

4. I couldn't find an answer on network density regulations at European scale. The reader can be curious if the number of monitoring sensors are arranged by some directives/regulations in EU e.g. Water Framework Directive etc. These aspects could make the content more fruitful then the current very technical classifications.

HESS REVIEW CHECK-LIST

1. Does the paper address relevant scientific questions within the scope of HESS? Yes, this is a review paper on sensor network design methods. I would, however, expect more insight on the general trend in reducing the number of the sensors in the world due to maintaining expenses. A particular example from Germany or another highly dense network country could be presented to the audience with more details. 2. Does the paper present novel concepts, ideas, tools, or data?

Yes, this is an extensive review on new design ideas, methods and concepts. 3. Are substantial conclusions reached?

Yes, especially inclusion of the hydrological models for network design is one of the important conclusions. 4. Are the scientific methods and assumptions valid and clearly outlined?

Yes, the authors explained the methods clearly. 5. Are the results sufficient to support the interpretations and conclusions?

Yes 6. Is the description of experiments and calculations sufficiently complete and precise to allow their reproduction by fellow scientists (traceability of results)?

This is a review paper. 7. Do the authors give proper credit to related work and clearly indicate their own new/original contribution?

Yes 8. Does the title clearly reflect the contents of the paper?

Yes 9. Does the abstract provide a concise and complete summary?

Yes 10. Is the overall presentation well structured and clear?

Yes but sub-titles should be better organized. 11. Is the language fluent and precise?

Yes 12. Are mathematical formulae, symbols, abbreviations, and units correctly defined and used?

Yes 13. Should any parts of the paper (text, formulae, figures, tables) be clarified, reduced, combined, or eliminated?

No 14. Are the number and quality of references appropriate?

Yes, enough 15. Is the amount and quality of supplementary material appropriate?

N/A

---

## Referee Comment (RC2) · Anonymous Referee #2 · 10 Oct 2016

This article presents a review of methodologies to address the design of sensor networks in hydrology and water management. The topic of the review is timely and certainly of interest to hydrologists and practitioners. However, the Authors should consider the following comments to improve on the overall clarity of the manuscript.

1) The manuscript language should be considerably improved. Please avoid typos and reword extensively to better clarify concepts.

2) Section 3 should be improved through a clear and simple explanation of underlying mathematical concepts and by adding representative case studies. Also, rather than listing applications, the Authors should provide comments on pros and cons for each approach, thus guiding the reader toward the selection of a suitable technique. Some-

times I found it difficult to follow the text as concepts were not properly connected. Few comments are devoted to Table 2 and to the Conclusions and recommendations.

3) Section 6 is poorly related to the others and its title is not sufficiently informative. I suggest Sections 5 and 6 are merged into a more comprehensive Discussion.

4) What is the relevance of the topic? I am sure of the importance of the subject but the Authors could better emphasize through key cases why the design of sensor networks is crucial and what major issues engineers/researchers may face in their definition.

HESS REVIEW CHECKLIST

1. Does the paper address relevant scientific questions within the scope of HESS?

Yes, the design of sensor networks is crucial in Hydrology and Earth System Sciences.

2. Does the paper present novel concepts, ideas, tools, or data?

Although this is a review paper, a novel framework to tackle sensor network design is presented.

3. Are substantial conclusions reached?

Yes but they need to be better commented and pointed out.

4. Are the scientific methods and assumptions valid and clearly outlined?

Yes but text should be improved.

5. Are the results sufficient to support the interpretations and conclusions?

No, methods should be better commented and case studies provided to support the suitability of the proposed framework.

6. Is the description of experiments and calculations sufficiently complete and precise to allow their reproduction by fellow scientists (traceability of results)?

Not applicable to a review paper.

7. Do the authors give proper credit to related work and clearly indicate their own new/original contribution?

Yes

8. Does the title clearly reflect the contents of the paper?

Partially, I'd expected more applications were more clearly reported.

9. Does the abstract provide a concise and complete summary?

Yes but it could be improved: the concept of using the performance of hydrological simulation of discharge as design criteria is not properly linked to the previous sentence.

10. Is the overall presentation well structured and clear?

The structure of the paper is fine but text needs improvement.

11. Is the language fluent and precise?

No, it needs considerable rewording.

12. Are mathematical formulae, symbols, abbreviations, and units correctly defined and used?

Not all formulas are clear. For instance, the explanation of terms in Eq. 16 (lines 405-407) is very confusing.

13. Should any parts of the paper (text, formulae, figures, tables) be clarified, reduced, combined, or eliminated?

Yes, text should be thoroughly clarified and typos avoided.

14. Are the number and quality of references appropriate?

Yes.

15. Is the amount and quality of supplementary material appropriate?

Not applicable.

---

## Referee Comment (RC3) · Anonymous Referee #3 · 24 Oct 2016

**General comments**

The manuscript presents a review of the existing methods for network sensor design for hydro-logical purposes. Moreover, in the introduction, the authors denote the lack of a unified methodology for network sensor design and, in the last paragraph, they propose a general procedure to fill this gap.

I personally have only few comments and I would suggest the publication of the paper, provided that the authors extend the text keeping in mind the following comments:

- I agree with the other two reviewers that a general overview about the network sensor densities at global or continental scale is missing. I would suggest to support these considerations with tables or maps to show some relevant characteristics of the networks. In case this is not possible because of the lack of data, I would suggest to add some study cases or examples that might be useful for decision-makers. This would trigger considerations for stakeholders about any actions to be undertaken and to provide answers to questions like "Under which circumstances should I re-evaluate my sensors networks? Should I improve, reduce or relocate sensors?"

- Some considerations about the advantages and disadvantages of the various methods for network sensor evaluation is missing. For example fractal approach methods suffer from the fact that they consider the sensors located in a two-dimensional space, ie not considering the elevation. On the contrary, orography might play an important role in the location of the precipitation maxima, thus fractal methods should be employed only in relatively flat areas. Another example where advantages and disadvantages might be relevant is the case of the methods based on expert judgment since these methods are, by definition, biased because of the expert.

- Since the method proposed in Section 5 is the novel concept introduced in the paper, I would appreciate an application of the method in a real case (for example a case when the optimal criteria are met to exit the loop and another case when they're not met). This would help the readers to conduct their own experiments based on this new tool.

**Specific comments**
The numbering of the Sections is sometimes confusing, I would suggest to simplify it (eg reducing the sub-sections) to get the text more smoothly. For example the Section 4 is very meager and I would merge it with another section (perhaps the last one?)

**Technical corrections**

- Please cite correctly the paper by Capecchi et al 2012 (not Cappechi et al 2011) and change the text accordingly

- Eq 13: The definition of joint entropy is not well explained for a non-expert. "max" in the right hand side of the formula is not clear, the dots "..." are not clear

- Eq 14: "m" stands for? "H" stands for? Please specify

- Since I'm not a native English speaker, I have no issues on the language. Anyway some typos are found; here some examples:

  – pag 16, line 531: "Heaviside function" with the capital letter
  – Figure 6, conditional block (7): "Is it..." instead of "Is It..."
  – Figure 6, conditional block (9): "Is it..." instead of "It is..."

**HESS review checklist**

- Does the paper address relevant scientific questions within the scope of HESS? Yes

- Does the paper present novel concepts, ideas, tools, or data? Yes, the unified methodology for optimal sensor design presented in Section 5 is a novel tool.

- Are substantial conclusions reached? Yes

- Are the scientific methods and assumptions valid and clearly outlined? Yes, but the review of the existing methods is sometimes confusing in terms of classification and Sections numbering

- Are the results sufficient to support the interpretations and conclusions? As stated previously in the General Comments, I would present a practical demonstration of the method described in section 5, to support the conclusions

- Is the description of experiments and calculations sufficiently complete and precise to allow their reproduction by fellow scientists (traceability of results)? Even if this is a review paper, a novel method is introduced (in Section5). According to my opinion a practical example of the method should be shown to help its reproduction by the readers

- Do the authors give proper credit to related work and clearly indicate their own new/original contribution? Yes they do

- Does the title clearly reflect the contents of the paper? Yes

- Does the abstract provide a concise and complete summary? Yes

- Is the overall presentation well structured and clear? I would re-organize the Sections numbering

- Is the language fluent and precise? Not applicable

- Are mathematical formulae, symbols, abbreviations, and units correctly defined and used? Yes, minor further details are needed in some equations (see above)

- Should any parts of the paper (text, formulae, figures, tables) be clarified, reduced, combined, or eliminated? (see above)

- Are the number and quality of references appropriate? Yes

- Is the amount and quality of supplementary material appropriate? Not applicable

---

## Author Comment (AC3) · 7 Dec 2016

» The manuscript presents a review of the existing methods for network sensor design for hydrological purposes. Moreover, in the introduction, the authors denote the lack of a unified methodology for network sensor design and, in the last paragraph, they propose a general procedure to fill this gap. I personally have only few comments and I would suggest the publication of the paper, provided that the authors extend the text keeping in mind the following comments: »

REPLY. We thank the reviewer for the precise and relevant comments. These comments have helped us to improve the manuscript.

» I agree with the other two reviewers that a general overview about the network sensor densities at global or continental scale is missing. I would suggest to support these considerations with tables or maps to show some relevant characteristics of the networks. In case this is not possible because of the lack of data, I would suggest to add some study cases or examples that might be useful for decision-makers. This would trigger considerations for stakeholders about any actions to be undertaken and to provide answers to questions like "Under which circumstances should I re-evaluate my sensors networks? Should I improve, reduce or relocate sensors?" »

REPLY. These comments were mainly pointed by Reviewer 1, and we replicate our reply to him/her in the following lines. We agree that practitioners may be interested in country-wise practices of hydrometric network expansion or modification. As the essence of the manuscript is to review the available mathematical methods to make such network expansions/modifications optimal, the connection to practical applications appeared weak.

In order to address the reviewer's comment, we have included references to country-scale network density, where the reader can find more detailed information (page 1, 31-34). We have also added statements to clarify that the optimal density of the network is case-specific (p3, 91-99), pointing out that practices in optimal monitoring network design would be, per-se, another in-depth study. We have framed these ideas in the new version of the paper without jeopardising its main focus. Also, main considerations about the selection of the appropriate number of gauges in the measurement-based methods are highlighted. In the new version of the manuscript we added the following text:

-> "Design of rainfall and streamflow sensor networks depends to a large extent on the scale of the processes to be monitored, and the objectives to address (TNO 1986, Loucks et al. 2005). Therefore, the temporal and spatial resolution of the measurements are driven by the measurement objectives. For example, information for long-term planning does not require the same level of temporal resolution as for operational

hydrology WMO (2009). On the global and country scale, sensor networks are commonly used for climate studies and trend detection (Cihlar et al. 2000, Grabs and Thomas 2002, WMO 2009, Environment Canada 2010, Marsh 2010, Whitfield et al. 2012). This is also supported by the National Climate Reference Networks (WMO 2009). On a regional or catchment-scale, applications require careful selection of monitoring stations, since water resources planning and management decisions, such as operational hydrology and water allocation, require different temporal and spatial resolution data."

-> "The sensor network design can also be seen from an economic perspective (Loucks et al. 2005). In most cases, the main limitation in the deployment of sensor networks is related to cost, being the main driver for the reduction scenarios. The valuation between the costs of the sensor networks and the cost of lack of information is not usually considered, because the assessment of the consequences of decisions is made a-posteriori (Alfonso et al. 2016). In most studies, it is seen that the improvement of information content metrics (e.g., entropy, uncertainty reduction, among others) is marginal as the number of extra sensors increases (Pardo-Iguzquiza 1998, Dong et al. 2006, Ridolfi et al. 2011), and thus the selection of the correct density can be based on a threshold in the increase in accuracy. However, in many practical applications, the number of available stations may be defined by budget limitations. Therefore, the optimal density of a sensor network is strictly case-specific (WMO 2008)."

To address the reviewer's particular comment on the sensor network re-evaluation, we have added more references to support our statement that it should be made on a regular basis. Considerations of the frequency of this re-evaluation are driven by the changes in the monitoring objectives, the available observation methods, budget restrictions and changes in the observed variable, among others (highlighted in section 1.1 p-4.), and, as one can imagine, these aspects are totally case-dependent.

The questions the reviewer is suggesting, like "Under which circumstances should I re-evaluate my sensors networks?", and "Should I improve, reduce or relocate sensors?" are indeed very important and we believe they should be addressed in a separate manuscript. From a review point of view, considerations of the frequency of the re-evaluation are driven by the changes in the monitoring objectives, the available observation methods, budget restrictions and changes in the observed variable. These considerations are highlighted in section 1.1 p-4.

» Some considerations about the advantages and disadvantages of the various methods for network sensor evaluation is missing. For example fractal approach methods suffer from the fact that they consider the sensors located in a two dimensional space, ie not considering the elevation. On the contrary, orography might play an important role in the location of the precipitation maxima, thus fractal methods should be employed only in relatively flat areas. Another example where advantages and disadvantages might be relevant is the case of the methods based on expert judgment since these methods are, by definition, biased because of the expert. »

REPLY. Indeed, highlighting advantages and disadvantages of different design methods provide a reference to the readers towards the selection of one method over another. This is a very good point, so we have added Table 3 presenting advantages and disadvantages of the different design methods. Table 3 can be found in the attachments of this reply.

» Since the method proposed in Section 5 is the novel concept introduced in the paper, I would appreciate an application of the method in a real case (for example a case when the optimal criteria are met to exit the loop and another case when they're not met). This would help the readers to conduct their own experiments based on this new tool. »

REPLY. We agree with the reviewer that presenting an example application of the proposed design methodology may be of value to the reader. Although this is the ongoing research, we find it too difficult to add it here, as it may compromise the scope and length of the paper. We would like to keep it as a review paper, with a proposed framework. We understand that proposing a framework in a review paper may outreach its limits, but considering that this methodology is implicitly addressed in many of the references, we identified it as an opportunity.

» Specific comments. The numbering of the Sections is sometimes confusing, I would suggest to simplify it (eg reducing the sub-sections) to get the text more smoothly. For example the Section 4 is very meager and I would merge it with another section (perhaps the last one?) »

REPLY. Thank you for the suggestion. We have simplified the paper structure by removing section 5, and merging its content in section 6. Additionally, we expand section 4 with Table 3. Table 3 can be found in the attachments of this reply.

» Technical corrections C2 Please cite correctly the paper by Capecchi et al 2012 (not Cappechi et al 2011) and change the text accordingly »

REPLY. We regret this mistake. It has been corrected.

» Eq 13: The definition of joint entropy is not well explained for a non-expert. "max" in the right hand side of the formula is not clear, the dots ". . . " are not clear »

REPLY. The formulas have been clarified.

» Eq 14: "m" stands for? "H" stands for? Please specify »

REPLY. The formulas have been clarified.

» Since I'm not a native English speaker, I have no issues on the language. Anyway some typos are found; here some examples: – pag 16, line 531: "Heaviside function" with the capital letter – Figure 6, conditional block (7): "Is it..." instead of "Is It..." – Figure 6, conditional block (9): "Is it..." instead of "It is..." »

REPLY. A complete revision of the paper has been undertaken to address the language issues.

[Figure]

1083 **Table 3 Advantages and disadvantages of sensor network design methods**

1084

| | Advantages | Disadvantages |
|---|---|---|
| **Statistics-based** | | |
| Interpolation variance | Useful to assess data scarce areas

No event-driven

Minimise uncertainty in spatial distribution of measured variable | Heavily rely on the characterisation of the covariance structure

No relationship with final measurement objective |
| Cross-correlation | Useful for detecting redundant stations

Computationally inexpensive | Augmentation not possible without additional assumptions

Limited to linear dependency between stations |
| Model error | Has direct relationship with the measurement objectives | Biased towards current measurement objectives

Biased towards model and error metrics |
| **Information Theory** | | |
| Entropy | Assess non-linear relationship between variables

Unbiased estimation of network performance | Formal form is computationally intensive

Quantising (binning) of continuous variables lead to different results

Optimal networks are usually sparse

Difficult to benchmark

Data intensive |
| Mutual information | Idem | Idem |
| **Expert recommendations** | | |
| Physiographic components | Well understood

Functional for heterogeneous catchments with few available measurements

Useful at country/continental level | Not useful for homogeneous catchments

No quantitative measure of network accuracy |
| Practical case-specific considerations | No previous measurements are required

Useful to observe specific variables | Biased towards expert

Collected data does not influence selection

Biased towards current data requirements |
| User survey | Pragmatic

Cost-efficient | Extensive user identification

Biased towards current data requirements |
| **Other methods** | | |
| Value of information | Provides a full economical assessment | Hard to quantify

Usually decisions are made with available information

Biased towards a rational decision model |
| Fractal characterisation | Efficient for large networks

Does not require data collection | Not suitable for small networks or catchments

Does not consider topographic or orographic influence |
| Network theory | Provides insight in interconnected networks | Not useful for augmentation purposes

Data intensive |

1085

35

**Fig. 1.**

---

## Author Response (AR1)

**Reply to Reviewer #1**

> » The authors present an overall picture on hydrometric network design methods and approaches to increase or reduce sensor density using different methods e.g. expert opinions and hydrologic models. They also classify these methods and present an optimal network design using complementary rainfall-runoff model performance. The use of hydrologic model makes sense as the products of the sensors are usually used by the hydrologic models. This review paper addresses an interesting topic. However, the presentation of the cases needs some more details on country scale applications as listed below. What are the practices in very densely monitored countries (e.g. Germany) and data scarce ones (e.g. Poland, Spain and Turkey). Also what is the optimum C1 level of network density.
>
> Overall, major revision is recommended for the the manuscript. »

**REPLY.** We thank the reviewer for the valuable contributions. This helped us with improving its quality, and also to address some points that could have been clearer or that were not considered with the adequate level of detail.

We agree with the reviewer that practitioners may be interested in country-wise practices of hydrometric network expansion or modification. As the essence of the manuscript is to review the available mathematical methods to make such network expansions/modifications optimal, the connection to practical applications appeared weak.

In order to address the reviewer's comment we have included references to country scale network density, where the reader can find more detailed information (page 1, 31- 40). We have also added statements to clarify that the optimal density of the network is case-specific (p3, 91-99), pointing out that practices in optimal monitoring network design would be, per-se, another in-depth study. We have framed these ideas in the new version of the paper without jeopardizing its main focus. Also, main considerations about the selection of the appropriate number of gauges in the measurement-based methods are highlighted. In the new version of manuscript we added the following text:

"Design of rainfall and streamflow sensor networks depends to a large extent on the scale of the processes to be monitored, and the objectives to address (TNO 1986, Loucks et al. 2005). Therefore, the temporal and spatial resolution of the measurements are driven by the measurement objectives. For example, information for long term planning does not require the same level of temporal resolution as for operational hydrology WMO (2009). On the global and country scale, sensor networks are commonly used for climate studies and trend detection (Cihlar et al. 2000, Grabs and Thomas 2002, WMO 2009, Environment Canada 2010, Marsh 2010, Whitfield et al. 2012). This is also supported by the National Climate Reference Networks (WMO C2 2009). On a regional or catchment-scale, applications require careful selection of monitoring stations, since water resources planning and management decisions, such as operational hydrology and water allocation, require different temporal and spatial resolution data. "

(for clarity, this section was slightly reworded as in p1 31-40)

*The design of rainfall and streamflow sensor networks depends to a large extent on the scale of the processes to be monitored and the objectives to address (TNO 1986, Loucks et al. 2005). Therefore, the temporal and spatial resolution of measurements are driven by the measurement objectives. For example, information for long-term planning does not require the same level of temporal resolution as for operational hydrology (WMO 2009, Dent 2012). On the global and country scale, sensor networks are commonly used for climate studies and trend detection (Cihlar et al. 2000, Grabs and Thomas 2002, WMO 2009, Environment Canada 2010, Marsh 2010, Whitfield et al. 2012), and denoted as National Climate Reference Networks (WMO 2009). On a regional or catchment-scale, applications require careful selection of monitoring stations, since water resources planning and management decisions, such as operational hydrology and water allocation, require high temporal and spatial resolution data (Dent 2012).*

"The sensor network design can also be seen from an economic perspective (Loucks et al. 2005). In most cases, the main limitation in the deployment of sensor networks is related to cost, being the main driver for the reduction scenarios. The valuation between the costs of the sensor networks and the cost of the lack of information is not usually considered, because the assessment of the consequences of decisions is made a-posteriori (Alfonso et al. 2016). In most studies, it is seen that the improvement of information content metrics (e.g., entropy, uncertainty reduction, among others) is marginal as the number of extra sensors increases (Pardo-Iguzquiza 1998, Dong et al. 2006, Ridolfi et al. 2011), and thus the selection of the correct density can be based on a threshold in the increase in accuracy. However, in many practical applications the number of available stations may be defined by budget limitations. Therefore, the optimal density of a sensor network is strictly case-specific (WMO 2008)."

(for clarity, this section was slightly reworded as in p3 97-106)

*The sensor network design can also be seen from an economic perspective (Loucks et al. 2005). In most cases, the main limitation in the deployment of sensor networks is related to costs, being sometimes the main driver of decisions related to reduction of the monitoring networks. The valuation between the costs of the sensor networks and the cost of having insufficient information is not usually considered, because the assessment of the consequences of decisions is made a-posteriori (Loucks et al. 2005, Alfonso et al. 2016). In most studies, it is seen that the improvement of information content metrics (e.g., entropy, uncertainty reduction, among others) is marginal as the number of extra sensors increases (Pardo-Iguzquiza 1998, Dong et al. 2006, Ridolfi et al. 2011), and thus the selection of the adequate number of sensors can be based on a threshold in the rate of increment in the objective function. However, in many practical applications the number of available sensors may be defined by budget limitations. Therefore, the optimal number of sensors in a network is strictly case-specific (WMO 2008).*

> » Specific Comments: 1. Title: Rainfall and streamflow sensor network design: a review of applications, classification, and a proposed framework Recommended title: Review of precipitation and streamflow sensor network design methods from hydrologic modeling perspective. »

**REPLY.** It is interesting that we suggested a similar title when we submitted this paper for the first time. During the first round of reviews, we found that the concept of hydrological modelling implied the inclusion of groundwater processes which are not included in our review. Therefore, we decided to avoid the term hydrological modelling, and try to manage readers' expectations in the title including only rainfall-runoff processes. We hope that the reviewer finds this decision adequate.

> » 2. Section/subsection titles should be reorganized in a clear way. For example subC3 section 3.3.2 Methods based on expert judgement and 3.3 Methods based on expert recommendations are similar and confusing. »

**REPLY.** We totally agree. We have renamed the methods in section 3.3.2 as 'Practical case-specific considerations', as we believe this better reflects the content. Additionally, section 5 (opportunities) has been removed and merged into the section Conclusions and Recommendations.

> » 3. In most of the European countries (e.g. Denmark and Germany) or even in USGS, the number of rainfall/streamflow sensors/stations is decreasing due to maintenance costs and use of radar data. I would expect to read some more insight on specific examples about sensor density and the country based approaches. Compare, for example, Spain/Poland and Germany from network density aspect to indicate an optimum approach. Now the content is very technical and dry for the reader. »

**REPLY.** Indeed, we agree that the practices within countries are different, and that there is a clear progress in monitoring technologies, such as radars and remote sensors. Although we believe that making the comparisons suggested by the reviewer would expand the current objective of our manuscript, we think that reviewing the current practices and monitoring plans of different authorities will beat the focus of our discussion. For this reason we have added a paragraph in this regard, in which the following useful references for the interested readers are included.

- Cihlar, J., W. Grabs, J. Landwehr. Establishment of a hydrological observation network for climate. Report of the GCOS/GTOS/HWRP expert meeting. Report GTOS 26. Geisenheim, Germany. WMO. 2000.
- EC. EU Water Framework Directive. Directive 2000/60/EC of the European Parliament and of the Council of 23 October 2000 establishing a framework for Community action in the field of water policy. European Commission. 2000.
- Grabs, W. and A. R. Thomas. Report of the GCOS/GTOS/HWRP expert meeting on C4 the implementation of a global terrestrial network – hydrology (GTN-H). Report GCOS 71, GTOS 29. Koblenz, Germany. WMO. 2001.
- WMO. Guide to hydrological practices. Volume II: Management of water resources and application of hydrological practices. WMO 168, 6th ed. 2009.
- Environment Canada. Audit of the national hydrometric program. 2010.
- Marsh, T. The UK Benchmark network – Designation, evolution and application. 10th symposium on stochastic hydraulics and 5th international conference on water resources and environment research. Quebec, Canada. 2010.
- Dent, J. E. Climate and meteorological information requirements for water management: A review of issues. WMO 1094. 2012.
- Withfield, P. H., D. H. Burn, J. Hannaford, H. Higgins, G. A. Hodgkins, T. Marsh and U. Looser. Reference hydrologic networks I. The status and potential future dierctions of national reference hydrologic networks for detecting trends. Hydrological Sciences Journal 57 (8), 1562 - 1579. doi:10.1080/02626667.2012.728706. 2012.

> » 4. I couldn't find an answer on network density regulations at European scale. The reader can be curious if the number of monitoring sensors are arranged by some directives/regulations in EU e.g. Water Framework Directive etc. These aspects could make the content more fruitful then the current very technical classifications. »

**REPLY.** Indeed, it is a relevant point to address. Most of the regulations consider monitoring necessities to meet a given observation objective, instead of defining (or suggesting) particular network densities. For example, the EU Water Framework Directive Article 8, states that "Member States shall ensure the establishment of programmes for the monitoring of water status in order to establish a coherent and comprehensive overview of water status within each river basin district", and only stipulates that technical specifications should be in accordance with a regulatory committee.

Other entities such as the USGS and Environment Canada do not outline regulations, C5 but monitoring plans. These are re-evaluated, in function of the monitoring objectives and budget limitations. Only WMO provides minimum density recommendations, as presented in the paper. We have extended the text pointing this out (p3 87-89):

"*Consequently, regulations regarding monitoring activities are not often strict in terms of station density, but in the suitability of data to provide information about the status of the water system (EC 2000, EPA 2002).*"

- EC. EU Water Framework Directive. Directive 2000/60/EC of the European Parliament and of the Council of 23 October 2000 establishing a framework for Community action in the field of water policy. European Commission. 2000.
- EPA. Guidance on choosing a sampling design for environmental data collection, EPA. US Environmental Protection Agency. 2002.

**Reply to Reviewer #2**

> » This article presents a review of methodologies to address the design of sensor networks in hydrology and water management. The topic of the review is timely and certainly of interest to hydrologists and practitioners. However, the Authors should consider the following comments to improve on the overall clarity of the manuscript. »

**REPLY.** We appreciate the thoughtful comments of the reviewer, and its constructive approach to improving the clarity and reach of this paper. The particular comments are addressed below. » 1) The manuscript language should be considerably improved. Please avoid typos C1 and reword extensively to better clarify concepts. » REPLY. We agree. The paper had a complete re-revision to improve language and clarity.

> » 2) Section 3 should be improved through a clear and simple explanation of underlying mathematical concepts and by adding representative case studies. Also, rather than listing applications, the Authors should provide comments on pros and cons for each approach, thus guiding the reader toward the selection of a suitable technique. Sometimes I found it difficult to follow the text as concepts were not properly connected. Few comments are devoted to Table 2 and to the Conclusions and recommendations. »

**REPLY.** This comment has triggered several changes in the manuscript, as Section 3 is one of the core sections of the paper. Indeed, Table 2 was extended to consider some relevant cases where the methods described in Section 3 are applied, thus guiding the reader into selected in-depth material. Additionally, and we thank the reviewer for the idea, a new table (Table 3) has been added to highlight advantages and disadvantages of the different methods. The new tables 2 and 3 are provided as an attachment to this reply.

> » 3) Section 6 is poorly related to the others and its title is not sufficiently informative. I suggest Sections 5 and 6 are merged into a more comprehensive Discussion. »

**REPLY.** We totally agree. We have merged Section 5 and 6.

> » 4) What is the relevance of the topic? I am sure of the importance of the subject but the Authors could better emphasize through key cases why the design of sensor networks is crucial and what major issues engineers/researchers may face in their definition. »

**REPLY.** We agree with the reviewer on highlighting the importance of sensor network design may help the paper reach a wider audience. However, we are concerned about doing it through case studies, as the context would necessarily change the focus of C2 the paper towards case-specific design practices or regulations. We therefore suggest the following compromise: we clarify the scope of the paper, and add a paragraph with references to literature (mostly reports) where the interested reader can find more information.

[revised manuscript text omitted]

Data intensive |

**Reply to Reviewer #3**

> » The manuscript presents a review of the existing methods for network sensor design for hydrological purposes. Moreover, in the introduction, the authors denote the lack of a unified methodology for network sensor design and, in the last paragraph, they propose a general procedure to fill this gap. I personally have only few comments and I would suggest the publication of the paper, provided that the authors extend the text keeping in mind the following comments: »

**REPLY.** We thank the reviewer for the precise and relevant comments. These comments have helped us to improve the manuscript.

> » I agree with the other two reviewers that a general overview about the network sensor densities at global or continental scale is missing. I would suggest to support these considerations with tables or maps to show some relevant characteristics of the networks. In case this is not possible because of the lack of data, I would suggest to add some study cases or examples that might be useful for decision-makers. This would trigger considerations for stakeholders about any actions to be undertaken and to provide answers to questions like "Under which circumstances should I re-evaluate my sensors networks? Should I improve, reduce or relocate sensors?" »

**REPLY.** These comments were mainly pointed by Reviewer 1, and we replicate our reply to him/her in the following lines. We agree that practitioners may be interested in country-wise practices of hydrometric network expansion or modification. As the essence of the manuscript is to review the available mathematical methods to make such network expansions/modifications optimal, the connection to practical applications appeared weak.

In order to address the reviewer's comment, we have included references to country-scale network density, where the reader can find more detailed information (page 1, 31- 40). We have also added statements to clarify that the optimal density of the network is case-specific (p3, 91-99), pointing out that practices in optimal monitoring network design would be, per-se, another in-depth study. We have framed these ideas in the new version of the paper without jeopardising its main focus. Also, main considerations about the selection of the appropriate number of gauges in the measurement-based methods are highlighted. In the new version of the manuscript we added the following text:

"Design of rainfall and streamflow sensor networks depends to a large extent on the scale of the processes to be monitored, and the objectives to address (TNO 1986, Loucks et al. 2005). Therefore, the temporal and spatial resolution of the measurements are driven by the measurement objectives. For example, information for long-term planning does not require the same level of temporal resolution as for operational hydrology WMO (2009). On the global and country scale, sensor networks are commonly used for climate studies and trend detection (Cihlar et al. 2000, Grabs and Thomas 2002, WMO 2009, Environment Canada 2010, Marsh 2010, Whitfield et al. 2012). This is also supported by the National Climate Reference Networks (WMO 2009). On a regional or catchment-scale, applications require careful selection of monitoring stations, since water resources planning and management decisions, such as operational hydrology and water allocation, require different temporal and spatial resolution data."

(for clarity, this section was slightly reworded as in p1 31-40)

*The design of rainfall and streamflow sensor networks depends to a large extent on the scale of the processes to be monitored and the objectives to address (TNO 1986, Loucks et al. 2005). Therefore, the temporal and spatial resolution of measurements are driven by the measurement objectives. For example, information for long-term planning does not require the same level of temporal resolution as for operational hydrology (WMO 2009, Dent 2012). On the global and country scale, sensor networks are commonly used for climate studies and trend detection (Cihlar et al. 2000, Grabs and Thomas 2002, WMO 2009, Environment Canada 2010, Marsh 2010, Whitfield et al. 2012), and denoted as National Climate Reference Networks (WMO 2009). On a regional or*

*catchment-scale, applications require careful selection of monitoring stations, since water resources planning and management decisions, such as operational hydrology and water allocation, require high temporal and spatial resolution data (Dent 2012).*

"The sensor network design can also be seen from an economic perspective (Loucks et al. 2005). In most cases, the main limitation in the deployment of sensor networks is related to cost, being the main driver for the reduction scenarios. The valuation between the costs of the sensor networks and the cost of lack of information is not usually considered, because the assessment of the consequences of decisions is made a-posteriori (Alfonso et al. 2016). In most studies, it is seen that the improvement of information content metrics (e.g., entropy, uncertainty reduction, among others) is marginal as the number of extra sensors increases (Pardo-Iguzquiza 1998, Dong et al. 2006, Ridolfi et al. 2011), and thus the selection of the correct density can be based on a threshold in the increase in accuracy. However, in many practical applications, the number of available stations may be defined by budget limitations. Therefore, the optimal density of a sensor network is strictly case-specific (WMO 2008)."

(for clarity, this section was slightly reworded as in p3 97-106)

*The sensor network design can also be seen from an economic perspective (Loucks et al. 2005). In most cases, the main limitation in the deployment of sensor networks is related to costs, being sometimes the main driver of decisions related to reduction of the monitoring networks. The valuation between the costs of the sensor networks and the cost of having insufficient information is not usually considered, because the assessment of the consequences of decisions is made a-posteriori (Loucks et al. 2005, Alfonso et al. 2016). In most studies, it is seen that the improvement of information content metrics (e.g., entropy, uncertainty reduction, among others) is marginal as the number of extra sensors increases (Pardo-Iguzquiza 1998, Dong et al. 2006, Ridolfi et al. 2011), and thus the selection of the adequate number of sensors can be based on a threshold in the rate of increment in the objective function. However, in many practical applications the number of available sensors may be defined by budget limitations. Therefore, the optimal number of sensors in a network is strictly case-specific (WMO 2008).*

To address the reviewer's particular comment on the sensor network re-evaluation, we have added more references to support our statement that it should be made on a regular basis. Considerations of the frequency of this re-evaluation are driven by the changes in the monitoring objectives, the available observation methods, budget restrictions and changes in the observed variable, among others (highlighted in section 1.1), and, as one can imagine, these aspects are totally case-dependent.

The questions the reviewer is suggesting, like "Under which circumstances should I re-evaluate my sensors networks?", and "Should I improve, reduce or relocate sensors?" are indeed very important and we believe they should be addressed in a separate manuscript. From a review point of view, considerations of the frequency of the re-evaluation are driven by the changes in the monitoring objectives, the available observation methods, budget restrictions and changes in the observed variable. These considerations are highlighted in section 1.1.

> » Some considerations about the advantages and disadvantages of the various methods for network sensor evaluation is missing. For example fractal approach methods suffer from the fact that they consider the sensors located in a two dimensional space, ie not considering the elevation. On the contrary, orography might play an important role in the location of the precipitation maxima, thus fractal methods should be employed only in relatively flat areas. Another example where advantages and disadvantages might be relevant is the case of the methods based on expert judgment since these methods are, by definition, biased because of the expert. »

**REPLY.** Indeed, highlighting advantages and disadvantages of different design methods provide a reference to the readers towards the selection of one method over another. This is a very good point, so we have added Table 3 presenting advantages and disadvantages of the different design methods. Table 3 can be found in the attachments of this reply.

> » Since the method proposed in Section 5 is the novel concept introduced in the paper, I would appreciate an application of the method in a real case (for example a case when the optimal criteria are met to exit the loop and another case when they're not met). This would help the readers to conduct their own experiments based on this new tool. »

**REPLY.** We agree with the reviewer that presenting an example application of the proposed design methodology may be of value to the reader. Although this is the ongoing research, we find it too difficult to add it here, as it may compromise the scope and length of the paper. We would like to keep it as a review paper, with a proposed framework. We understand that proposing a framework in a review paper may outreach its limits, but considering that this methodology is implicitly addressed in many of the references, we identified it as an opportunity.

> » Specific comments. The numbering of the Sections is sometimes confusing, I would suggest to simplify it (eg reducing the sub-sections) to get the text more smoothly. For example the Section 4 is very meager and I would merge it with another section (perhaps the last one?) »

**REPLY.** Thank you for the suggestion. We have simplified the paper structure by removing section 5, and merging its content in section 6. Additionally, we expand section 4 with Table 3. Table 3 can be found in the attachments of this reply.

> » Technical corrections C2 Please cite correctly the paper by Capecchi et al 2012 (not Cappechi et al 2011) and change the text accordingly »

**REPLY.** We regret this mistake. It has been corrected.

> » Eq 13: The definition of joint entropy is not well explained for a non-expert. "max" in the right hand side of the formula is not clear, the dots ". . . " are not clear »

**REPLY.** The formulas have been clarified.

> » Eq 14: "m" stands for? "H" stands for? Please specify » REPLY. The formulas have been clarified. » Since I'm not a native English speaker, I have no issues on the language. Anyway some typos are found; here some examples: – pag 16, line 531: "Heaviside function" with the capital letter – Figure 6, conditional block (7): "Is it..." instead of "Is It..." – Figure 6, conditional block (9): "Is it..." instead of "It is..." »

**REPLY.** A complete revision of the paper has been undertaken to address the language issues.

[revised manuscript text omitted]

$$\max H(X_1, X_2, ..., X_n) = \max - \sum_{i=1}^{m} ... \sum_{j=1}^{n} p(x_{i1}, ... x_{jm}) \log p(x_{i1}, ... x_{jm}) H(X_1, X_2, ..., X_n)$$

$$= - \sum_{i=1}^{k} ... \sum_{j=1}^{m} p(x_{i1}, ... x_{jm}) \log p(x_{i1}, ... x_{jm})$$

( 13 )

Where $p(X)$ is the probability of the random variable $X$ to take thea discrete value $x_m$. As in many applications, $x_m X$

is a continuous value; the variable $X$which has to be discretised (quantised) into intervals before$(k, m)$ to calculate its entropy. The probabilities are calculated following frequency analysis, such that the probability of a variable $X$

to take a value in the interval $i,...,j$ which is defined by the calculation of number of times in which this value appear, divided by the (Joint) Entropycomplete length of the dataset. When calculating the entropy of more than one variable simultaneously (joint entropy), joint probabilities are used.

Krstanovich and Singh (1992) presented a concise work on rainfall network evaluation using Entropyentropy.

They used POME to obtain multivariate distributions to associate different dependencies between sensors, such as joint information and shared information, which was used later either reduce the network (in the case of high redundancy) or expand it (in the case of lack of common information).

Fuentes et al. (2007) proposed an Entropyentropy-utility criterion for environmental sampling, particularly suited for air-pollution monitoring. This approach considers Bayesian optimal sub-networks using an Entropyentropy framework, relying on the spatial correlation model. An interesting contribution of this work is the assumption of non-stationarity, contrary to traditional atmospheric studies, and relevant in the design of precipitation sensor networks.

The use of hydraulic 1D models and metrics of Entropyentropy have been used to select the adequate spacing between sensors for water level in canals and polder systems (Alfonso et al. 20142010a,b). This approach is based on the current conditions of the system, which makes it useful for operational purposes, but it does not necessarily support the modifications in the water system conditions or changes in the operation rules. Studies on the design of sensor networks using these methods are on the rise in the last years (Alfonso 2010, Alfonso et al. 2013, Ridolfi et al. (2013, Banik et al 2017).

Benefits of POME include the robustness of the description of the posterior probability distribution since it aims to define the less biassed outcome. This is because neither the models nor the measurements are completely certain.

Li et al. (2012) presented, as part of a multi-objective framework for sensor network optimisation, the criteria of maximum (Joint) Entropyjoint) entropy, as one of the objectives. Other studies in this direction have been presented by Lindley (1956), Caselton and Zidek (1984), Guttorp et al. (1993), Zidek et al. (2000), Yeh et al.

(2011) and Kang et al. (2014).

More recently, Samuel et al. (2013) and Coulibaly and Samuel (2014), proposed a mixed method involving regionalisation and dual Entropyentropy multi-objective optimisation (CRDEMO). This method), which is a step forward if compared to single-objective optimisation methods for sensor network design.

**3.2.2 Mutual information (trans-information) **

Mutual information is a measurement of the amount of information that a variable contains about another. This is measured as the *relative entropy* between the joint distribution and the product distribution (Cover and Thomas 2005). In the simplest expression (two variables), the mutual information can be defined as:

$$\min I(X_1, X_2, \ldots, X_n) = \min \sum_{i=1}^{m} \sum_{j=1}^{n} \frac{H(X_1, X_2, \ldots, X_n)}{p(x_{1,t}) p(x_{2,t}) \ldots p(x_{n,t})}$$

$$I(X_1, X_2) = H(X_1) + H(X_2) - H(X_1, X_2)$$

( 14 )

where $H(X_1)$ and $H(X_2)$ is the entropy of each of the variables, and $H(X_1, X_2)$ is the joint entropy between them. The extension of the mutual information for more than two variables should not only consider the joint entropy between them, but also the joint entropy between pairs of variables, leading to a significantly complex expression for the multivariate mutual information. Regarding this  issue, the multivariate mutual information can be addressed as a nested problem, such that:

$$I(X_1, X_2, \ldots, X_n) = I(X_1, X_2, \ldots, X_{n-1}) - I(X_1, X_2, \ldots, X_{n-1}|X_n)$$

( 15 )

[revised manuscript text omitted]

Data intensive |